# Small-scale heterogeneity of trace metals including REY in deep-sea sediments and pore waters of the Peru Basin, SE equatorial Pacific

Sophie A. L. Paul[1], Matthias Haeckel[2], Michael Bau[1], Rajina Bajracharya[1], Andrea Koschinsky[1]

[1]Department of Physics and Earth Sciences, Jacobs University Bremen, Bremen, 28759, Germany
[2]GEOMAR Helmholtz Centre for Ocean Research Kiel, Kiel, 24148, Germany

*Correspondence to*: Sophie A. L. Paul (s.paul@jacobs-university.de)

**Abstract.** Due to its remoteness, the deep-sea floor remains an understudied ecosystem of our planet. The patchiness of existing data sets makes it difficult to draw conclusions about processes that apply to a wider area. In our study we show how different settings and processes determine sediment heterogeneity on small spatial scales. We sampled solid phase and pore water from the upper 10 m of an approximately 7.4 x 13 km$^2$ large area in the Peru Basin, south-east equatorial Pacific Ocean, at 4100 m water depth. Samples were analyzed for trace metals including rare earth elements and yttrium (REY) as well as for particulate organic carbon (POC), $CaCO_3$, and nitrate. The analyses revealed a surprisingly high spatial small-scale heterogeneity of the deep-sea sediment composition. While some cores have the typical green layer from Fe(II) in the clay minerals, this layer is missing in other cores, i.e. showing a tan color associated with more Fe(III) in the clay minerals. This is due to varying organic carbon contents: nitrate is depleted at 2-3 m depth in cores with higher total organic carbon contents, but is present throughout cores with lower POC contents, thus inhibiting the Fe(III)-to-Fe(II) reduction pathway in organic matter degradation. REY show shale-normalized (SN) patterns similar to seawater with a relative enrichment of heavy REY over light REY, positive $La_{SN}$ anomaly, negative $Ce_{SN}$ anomaly, as well as positive $Y_{SN}$ anomaly and correlate with the Fe-rich clay layer and in some cores also with P. We, therefore, propose that Fe-rich clay minerals, such as nontronite, as well as phosphates are the REY-controlling phases in these sediments. Variability is also seen in dissolved Mn and Co concentrations between sites and within cores, which might be due to dissolving nodules in the suboxic sediment, as well as in concentration peaks of U, Mo, As, V, and Cu in two cores, which might be related to deposition of different material at lower lying areas or precipitation due to shifting redox-boundaries.

## 1 Introduction

### 1.1 Fragmentary data sets from the deep sea

The deep-sea floor below 1000 m covers approximately 60% of our planet's solid surface (Glover and Smith, 2003). Less than 0.01% has however been sampled and investigated in detail so far (Ramirez-Llodra et al., 2010), resulting in a scarce dataset. A recent study in the Clarion Clipperton Zone (CCZ) in the central equatorial Pacific found large-scale biogeochemical heterogeneity with respect to e.g., sedimentation rate, particulate organic carbon (POC) flux, POC contents, oxygen penetration depth, and thereby extension of the oxic and suboxic zones (Volz et al., 2018). The analyses, however, were based on one core per work area only, separated by hundreds of km.

Similarly, many studies in the past collected cores for pore-water and solid-phase geochemical analyses based on sparse sampling distribution and spread over large areas (e.g., Froelich et al., 1979; Klinkhammer, 1980; Toyoda and Masuda, 1991; Drodt et al., 1997; König et al., 1997, 1999; Haley et al., 2004; Schacht et al., 2010; Kim et al., 2012; Soyol-Erdene and Huh, 2013; Kon et al., 2014; Deng et al., 2017; Volz et al., 2018; Abbott et al., 2019).

Processes might, however, vary even on small spatial scales. For example, Mewes et al. (2014) showed small-scale biogeochemical pore-water variability in the German contract area for deep-sea mining in the CCZ. It remains to determine whether studies of a few isolated samples are representative for large areas of the deep sea or if these results are coincidental snapshots of a largely unknown, heterogeneous bigger picture.

**1.2 Previous work in the Peru Basin**

In contrast to most other deep-sea basins, the Peru Basin, located in the south-east central Pacific at approx. 4100 m water depth (Fig. 1), has been comparably well investigated including the geochemical composition of its sedimentary solid phase, pore water, and early diagenetic processes (Haeckel et al., 2001; König et al., 2001; Koschinsky, 2001; Koschinsky et al., 2001b, 2001a; Marchig et al., 2001; ; Stummeyer and Marchig, 2001; Paul et al., 2018). This is because it has been used as a study site for impacts of polymetallic nodule mining on the

abyssal environment in the 1980s and 1990s (Thiel and Schriever, 1990; Thiel, 2001). Polymetallic nodules are mineral precipitates of Mn oxides and Fe (oxyhydr)oxides that form around a nucleus, e.g. bone, rock or nodule fragments, from accretion of Mn oxides and Fe (oxyhydr)oxides from seawater and pore water (Hein and Koschinsky, 2014). Thus, with renewed scientific, industrial, and political interest in deep-sea mining, the Peru Basin has recently received attention again, which allowed for detailed biogeochemical investigations.

In 1989, a *DISturbance and reCOLonization experiment* (DISCOL) was started to investigate potential impacts of polymetallic nodule mining in the Peru Basin (Thiel and Schriever, 1990). The seafloor was plowed in a 11 km² large circular field, disturbing the upper decimeters of the surface sediment and removing the nodules from the surface (Thiel and Schriever, 1990). Geochemical investigations of nutrients, dissolved organic carbon (DOC),

amino acids, solid-phase and dissolved trace metals were conducted as part of the follow-up project ATESEPP in 1996 (Schriever et al., 1996). The geochemical work focused on the bioturbated surface layer, where impacts of polymetallic nodule mining are expected (Haeckel et al., 2001; Koschinsky, 2001; Koschinsky et al., 2001a, 2001b), whereas geochemical investigations of deeper sediment layers down to 10 m were only performed on five cores (with only one of them located in the DISCOL area) (Haeckel et al., 2001). Mineralogical investigations of

long cores were, however, conducted extensively (Weber et al., 1995, 2000; Marchig et al., 2001). As part of recent work in the MiningImpact project (https://jpio-miningimpact.geomar.de), the focus lay again on the surface sediments (Haffert et al., in review; Paul et al., 2018). To understand biogeochemical processes over longer time scales and to resolve more steps of the redox-zonation, the biogeochemical analysis of long sediment cores is crucial.

**1.3 Early diagenesis in the Peru Basin**

The Peru Basin is located at the southern border of the equatorial high-productivity zone (Weber et al., 2000), where it receives high inputs of particulate organic matter. As a consequence, POC contents are 0.5-1 wt.% and oxygen penetrates approx. 5-25 cm into the sediment (Haeckel et al., 2001; Paul et al., 2018). The oxic surface sediments are rich in Mn oxides and associated elements, giving this layer its dark brown color (Koschinsky, 2001;

Paul et al., 2018). Below the oxic zone, the sediment is suboxic and Mn oxides are reduced in the course of suboxic POC degradation, leaving the sediment with a tan color. The Fe(III)-to-Fe(II) redox boundary is assumed to occur where the sediment color changes from tan to green. The depth of the tan-green color change coincides with the $NO_3^-$ penetration depth as the color change typically indicates re-oxidation of Fe(II) to Fe(III) by $NO_3^-$ ( Lyle, 1983; Drodt et al., 1997; König et al., 1997, 1999). The Fe(II)/Fe(III) ratio changes from approx. 11/89 in the tan layer to 37/63 in the green layer (König et al., 1997). The first four steps of the typical redox sequence of marine sediments presented by Froelich et al. (1979) – oxygen, nitrate, Mn oxide, and Fe reduction – that develops from the energy gain of the electron acceptors utilized in the degradation of organic matter, are therefore visible here (König et al., 1999; Paul et al., 2018).

**1.4 Fe-rich clay minerals**

Sediments of the DISCOL area are mainly composed of siliceous and calcareous oozes and muds (Weber et al., 1995). The predominant clay minerals in the sediments are illite, kaolinite, and chlorite (of largely detrital origin) while smectites (authigenic clay minerals) – such as montmorillonite and nontronite – are present in smaller quantities (Fritsche et al., 2001; Marchig et al., 2001). In the Peru Basin, the concentration of nontronite and other authigenic clay minerals increases with increasing distance from the continent (Marchig et al., 2001). Nontronite is the Fe(III)-rich member of the smectite group (Murnane and Clague, 1983) and the structural Fe(III) in nontronite can be reduced reversibly ( Russell et al., 1979; Dong et al., 2009). With reduction, the color changes from yellowish to blue-green (Russell et al., 1979 and references therein; Lyle, 1983; Drodt et al., 1997; König et al., 1997). In contrast, Al-rich smectites darken from off-white to gray upon reduction (Lyle, 1983 and references therein).

**1.5 Rare earth elements and yttrium (REY)**

Rare earth elements and yttrium (REY) are frequently used to reconstruct physico-chemical environmental conditions and sediment provenance (e.g., Bau and Dulski, 1999; Bright et al., 2009). Yttrium is trivalent like the rare earth elements (REE) and of similar ionic size as Ho and therefore closely associated with the REE – then commonly called REY. The REE have slightly decreasing ionic radii with increasing atomic number which can lead to fractionation, resulting in distinct patterns in the shale-normalized (SN) plots, which can be used to differentiate REY-controlling phases and sedimentary processes. Fractionation can indicate particle-solution interactions in the marine environment, when for example Ce or Y are decoupled from their REY neighbors during redox cycling or hydrogenetic Mn and/or Fe (oxyhydr)oxide formation, respectively ( Bau et al., 1997, 1998; Bau, 1999). This is because of different surface complex stabilities between the individual REY (Bau et al., 1997). The subtle differences between complex stability constants are sufficient to lead to fractionation because of preferential scavenging or mobilization of the light REY (LREY; La-Nd), middle REY (MREY; Sm-Dy), or heavy REY (HREY; Y-Lu) (Cantrell and Byrne, 1987; Elderfield, 1988). Analyses of REY in sediments in other areas of the Pacific often found a REY association with Ca phosphates or Fe phases (e.g., Elderfield et al., 1981; Toyoda et al., 1990; Kashiwabara et al., 2018; Liao et al., 2019; Paul et al., 2019), the latter especially in areas of hydrothermal activity (Ruhlin and Owen, 1986; German et al., 1990). $REY_{SN}$ patterns of apatite pellets from Peru shelf sediments display heavy REY (HREY) enrichment as well as pronounced negative $Ce_{SN}$ anomalies and positive $Y_{SN}$ anomalies (Piper et al., 1988). Sediments from the Peru Basin that were interpreted to be hydrothermally influenced showed $REY_{SN}$ patterns similar to seawater (HREY enrichment, negative $Ce_{SN}$

anomaly, positive La$_{SN}$ anomaly) but Eu and Y were not reported and measured, respectively (Marchig et al., 1999). Europium can be used to identify a high-temperature hydrothermal influence on the sediment (Michard, 1989; German et al., 1990; Bau, 1991).

Clay minerals such as illite and kaolinite show flat REY patterns when normalized to Post Archean Australian Shale (PAAS), European Shale (EUS) or any other analogue of average upper continental crust material, due to their detrital origin (Cullers et al., 1975; Prudêncio et al., 1989; Marchig et al., 2001; Tostevin et al., 2016). Nontronite of hydrothermal origin displays seawater-like REY$_{SN}$ patterns except for a less pronounced Ce$_{SN}$ anomaly (Murnane and Clague, 1983; Alt, 1988) and sometimes a Eu$_{SN}$ anomaly (Mascarenhas-Pereira and Nath,
2010). To the best of our knowledge, no REY data from non-hydrothermal nontronite has been published yet.

**1.6 Research aim**

The sampling campaign of the *Joint Programming Initiative of Healthy and Productive Seas and Oceans pilot action "Ecological aspects of deep-sea mining" (MiningImpact;* https://jpio-miningimpact.geomar.de*)* conducted with RV SONNE in 2015 found that the sediments in the upper 10 mbsf are surprisingly heterogeneous in the
approx. 7.4 x 13 km$^2$ wide study area. Therefore, we aim to address the question: Which parameters show heterogeneity with respect to sediment composition and sedimentation input? To shed more light onto this small-scale regional variability, we investigated trace metal distributions in the solid phase and corresponding pore water to distinguish patterns and exceptions with respect to sediment layers, impacts of bathymetry, and early diagenetic processes. We consider such information on small-scale variability important for interpreting the
representativeness of individual sediment cores on which previous studies were often based. Here, we focus on parameters relevant for the description of the redox-zonation (POC, NO$_3^-$, Mn, Fe, and the Mn associated metals Co and Ni), REY and indicators for their controlling phases (P, Al, Fe), CaCO$_3$ and Ba for paleo-reconstructions, and redox sensitive elements such as U, Mo, As, V, as well as Cd and Cu.

**2 Methods**

**2.1 Sampling area and methods**

Samples were collected from seven gravity cores (GC) during RV SONNE cruise SO242/1 in 2015 to the Peru Basin (Greinert, 2015). A disturbance experiment mimicking nodule mining was conducted in this area in 1989 (DISCOL project), during which a circular area of approximately 11 km$^2$ was traversed with a plow harrow (Thiel and Schriever, 1990). The affected area is called the DISCOL experimental area (DEA), while undisturbed sites
around this area are reference areas. Three cores were sampled in reference sites (South, West, East) of this experimental set-up, which are spread around the DEA within ca. 80 m difference in water depth. Within the DEA, we sampled one slightly low-lying area (trough) as well as an area without nodules at the surface, corresponding to low acoustic backscatter intensity in the side-scan sonar images (black patch). In addition, one GC was taken inside an inactive small volcanic crater in close proximity to the DEA (Fig. 2, Table 1). Areas connected to the
deep-sea mining experimental sites are also listed in Table 1.

The plowing affected approximately the upper 20 cm of the sediment in the tracks and less in areas of resettled sediment, which was determined based on multicorer (MUC) data from the DISCOL area, including the sites

corresponding to the GCs presented here (Paul et al., 2018). This upper layer is often lost or disturbed during GC sampling so that the disturbance experiment should not affect the comparison of the GCs, regardless whether they were sampled in disturbed or undisturbed sites. As the GCs are not sampled with video guidance, it is unclear if a GC was taken exactly in a track or not; therefore, a comparison of disturbed and undisturbed sites is not possible

based on GCs.

## 2.2 Sediment and pore-water sampling

Once on deck, GCs were cut into 1 m sections and then divided into a working and an archive half. Working halves were instantly transported to the cold room (approx. 4°C), while the counterparts were stored as archive halves.

Samples were immediately collected to minimize contact with ambient air and thereby oxidation of reduced species in suboxic sections of the cores. After visual inspection, sediment was sampled in layers of different color, roughly one to two per meter, and transferred with plastic spoons into 50 mL acid pre-cleaned centrifuge tubes. Gravity core subsampling in ambient air is standard procedure and has been carried out regularly in previous studies (see e.g., Haeckel et al., 2001; Volz et al., 2018). Einstein-Smoluchowski informs us that diffusion will

carry solutes, such as $O_2$, only over a distance of 3 mm in 2 hours. Hence, our sampling after splitting the core is quick enough to ensure an almost pristine signal. Our experience with more sensitive variables, such as $H_2S$ and $Fe^{2+}$, supports this. The significant loss of dissolved constituents by oxidation is therefore not expected in the few hours of sampling, especially when sampling in low temperature conditions (for Mn(II) see e.g., Schnetger and Dellwig, 2012). Data for other redox-sensitive elements, e.g. U, Mo, V, As, compare well with pore-water data

from multicores from these sites, which were sampled in glove bags (Paul et al., 2018). Additionally, centrifuge tubes were completely filled to minimize the oxygen content during centrifugation. Samples were centrifuged at 3200 rpm/2061 xg for 40 minutes at 4°C to separate pore water from the solid phase. In a glove box with a steady stream of argon gas, pore water was then filtered through 0.2 µm cellulose acetate (CA) syringe filters, which had been cleaned previously with 0.1 M suprapure hydrochloric acid (HCl) and deionized water. In the past, sometimes

0.45 µm filters were used, e.g., in studies by Shaw et al.( 1990) and Beck et al. (2008)  to which we also compare the dissolved concentrations but this intercomparison is frequently done and no significant differences have been noticed so far. Pore-water samples were acidified with suprapure HCl (30%) using 1 µL for 1 mL of sample and kept cool until further analysis.

## 2.3 Chemical analyses

To determine bulk sediment metal concentrations, 100 mg of ground and oven-dried (105°C) sample was acid pressure digested in a PicoTrace DAS system at 220°C for 12 hours using 3 mL of perchloric acid ($HClO_4$, 70%, suprapure) and 3 mL hydrofluoric acid (HF, 38-40%, suprapure). Samples were evaporated and taken up in HCl (20-30%, suprapure) two times and at the end in 0.5 M nitric acid ($HNO_3$, suprapure) and 0.47 M HCl (suprapure). Some digested samples had small black particles left after the digestion and were filtered through 0.2 µm CA

filters prior to analyses. Method comparison with other geochemistry labs at the beginning of this project showed that the black particles do not affect the final results. Samples digested using the method above compared well with samples digested in a microwave digestion system using HCl, $HNO_3$, and HF and samples did not show black particles. For major elements, solutions of digested solid-phase samples were measured with ICP-OES (SpectroCiros SOP instrument) and for trace elements, including REY, with ICP-MS (Perkin Elmer Nexion 350x)

at Jacobs University Bremen. For pore-water analyses, the sample was first passed through an *apex Q* (ESI)connected to the ICP-MS. The desolvation nebulizer introduces the sample solution into a heated spray chamber and subsequently into a cooled condenser The *apex Q* thereby enhances sample introduction efficiency to decrease background noise and to increase sensitivity. Additionally, dissolved V, Mn, Co, Cu, As, and Mo were measured in kinetic energy discrimination mode using He gas to remove polyatomic interferences. The certified reference materials (CRM) MESS-3 and BHVO-2 were used for sediment and NASS-6, NASS-7, and SLEW-3 for pore-water samples ([www.nrc-cnrc.gc.ca](www.nrc-cnrc.gc.ca); crustal.usgs.gov). Accuracy and precision were determined based on averages of the CRMs from ICP-OES and ICP-MS runs.  Accuracy for Al in MESS-3 during ICP-OES measurements (n=13) was within 20% but has been known for too low Al values for some digestion methods (Roje, 2010). Data below the limit of quantification (LOQ) were excluded, except for pore-water As values of 84GC and 132GC due to good agreement of NASS-7 As data, which is in the same range as the sample concentrations. For detailed information about LOQ, accuracy, and method precision see Supplementary Material 1.

**2.4 Nitrate**

Nitrate was measured directly after sampling on board RV SONNE. Analyses followed standard procedures described by Grasshoff et al. (1999), using Cd for reduction to $NO_2^-$ and determining it as sulphanile-naphthylamide with a Hitachi UV/VIS spectrophotometer. Method precision was 3% and the limit of detection 2 µmol/L.

**2.5 Particulate organic carbon (POC) and $CaCO_3$**

Total carbon of freeze-dried, ground sediment was measured at the GEOMAR laboratories in Kiel with a Carlo-Erba NA-1500 Elemental Analyzer, analyzing $CO_2$ that was produced by flash combustion. To determine total organic carbon and $CaCO_3$, carbonate-bound carbon was removed with HCl from the sample prior to organic carbon measurement and the total inorganic carbon content was calculated from the difference between total carbon and organic carbon. It was then converted to $CaCO_3$ wt.%.

Detailed tables with data for major and trace elements as well as links to associated data sets of other pore-water and solid-phase parameters ($NO_3^-$, POC, $CaCO_3$) are available online at PANGAEA: [https://doi.org/10.1594/PANGAEA.903019](https://doi.org/10.1594/PANGAEA.903019)

**2.6 Depth correction for GCs and $CaCO_3$ correction**

Part of the semi-liquid surface sediments of the DISCOL area is typically lost from the GCs when placing the barrel horizontally on deck. Hence, the thickness of the lost sediment was estimated by comparison of various geochemical data (i.e. POC, $CaCO_3$, porosity, dissolved silicate) and core photos of the GCs with MUCs to derive true sediment depths of the samples. On average, between 10 and 30 cm were lost before sampling.

Solid-phase data (except Ca) is presented on a carbonate-free basis and was corrected for CaCO$_3$ due to high carbonate concentrations in some layers (Eq. (1)). Different sediment aliquots were taken for CaCO$_3$ and metal analyses and therefore the corrections were calculated using CaCO$_3$ data within an up to 15 cm range of mean metal sediment sample depth. Buried nodule data was not corrected for CaCO$_3$. For more details see https://doi.pangaea.de/10.1594/PANGAEA.903517.

$$[\text{concentration}_{\text{corrected}}] = \frac{[\text{concentration}]}{(100 - [\text{CaCO3 wt.\%}])} * 100 \tag{1}$$

**2.7 Reporting of REY data**

All REY patterns are normalized to PAAS, using REY data from McLennan (1989); normalization to EUS (Bau et al., 2018) or any other analogue of average upper crustal material provides similar REY$_{SN}$ patterns. Anomalies of REY in the SN patterns were calculated as described in Eq. (2). This equation calculates the ratio of e.g., Ce$_{SN}$/Ce$_{SN}$* which results in the value of the anomaly and helps to discern the extent of the respective anomaly. Calculation of Ce$_{SN}$ anomaly after Bau and Dulski, 1996a:

$$\frac{Ce}{Ce*} = \frac{Ce(SN)}{(0.5*La(SN) + 0.5*Pr(SN))} \tag{2}$$

**3 Results**

**3.1 Core descriptions**

The Mn-oxide-rich dark brown top layer was largely lost in all gravity cores, except for the core from *Small Crater* where 10 cm remained. In the *Reference West* core it was completely absent. Below, all cores have a light brown to grayish brown color (2.5Y5/2 or 6/2 on the Munsell color chart; de Stigter, 2015) until approx. 2-2.5 m, followed in four cores (i.e. *Reference South, DEA Black Patch, DEA Trough,* and *Reference East*) by a greenish gray color (5Y5/2, 5Y6/2; 5GY5/1 on the Munsell color chart; de Stigter, 2015)) to approx. 5-7 m depth. The cores of *Reference West, DEA West*, and *Small Crater* showed an olive color (2.5Y5/3 at around 1 m in the *DEA West* core and 2.5Y5/4 at around 1-2 m in the *Small Crater* core). At the bottom 2-2.5 m of all GCs mottled dark brown sediment (10YR4/3, 4/4 and 5/4 on the Munsell color chart; de Stigter, 2015) was found (Fig. 3).

The GCs of *Reference South* and *Reference West* recovered a nodule from the sediment surface, whereas buried nodules were found in the cores of *Reference West* at 458 cm*, *DEA Trough* at 387 cm*, 468 cm*, 564 cm, and 667 cm*, *Reference East* at 290 cm*, 346 cm, 747 cm and 870 cm, and *Small Crater* at 719 cm and 792 cm, the ones with an asterisk being analyzed as part of this study. Consequently, buried nodules exist below 290 cm in the DISCOL area. The dissolving nodules in *DEA Trough* at 468 cm, 564 cm and 667 cm, and at *Reference East* at 290 cm and 747 cm have brownish 'halos' around them in the green sediment. In *DEA Black Patch* at 497 cm and in *DEA Trough* at 585 cm, there are brown patches within the green sediment without a buried nodule being visible.

In *Reference East*, diffuse dark gray bands of approximately 1 cm thickness are found at depths of 229.5 cm, 236.5 cm and 330 cm. The dark gray bands are present again between 324 cm and 358 cm, from 386 cm to 402 cm and

510 cm to 518 cm depth (de Stigter, 2015). Between 476 cm and 500 cm, the gray bands extend vertically (de Stigter, 2015).

POC and nitrate are presented because they are important parameters when analyzing the redox-zonation of marine
sediments. POC contents in the sediment vary between approx. 0.5 and 0.8 wt.% in the upper layers and decrease with depth to approx. 0.1 to 0.4 wt.% (Fig. 4). Nitrate concentrations are 50-70 µmol/L in surface sediments and are depleted (<10 µmol/L) within the upper 2-3 m, except in cores *Reference South*, where $NO_3^-$ is depleted at ~6 m, and *Reference West* and *Small Crater*, where $NO_3^-$ remains at approx. 25 µmol/L throughout the core (Fig. 3).

**3.2 Ca, CaCO$_3$, Ba, Al, Fe, Mn and associated metals**

Calcium concentrations are around 1 wt.% throughout most of the sediment cores with increased concentrations of up to 15 wt.% between 150 and 500 cm as well as between 800 and 1000 cm (Fig. 5). Calcium carbonate concentrations are therefore also elevated in these depth ranges, with concentrations of up to 35 wt.%. Barium concentrations are generally between 0.5 and 1 wt.% in the upper 400 cm and increase downcore, except at *Small*
*Crater*, where concentrations are relatively constant (Fig. 5).

Aluminum concentrations decrease below 400 cm depth, most strongly at the western sites *Reference West* and *DEA West*. In these cores, concentrations of P, Cu, Mn, as well as metals associated with Mn, such as Ni, and Co, increase below 400 cm (Fig. 6). Iron displays a constant concentration of 3-4 wt.% down to 3-4 m. Further below,
Fe concentrations increase up to 7.5 wt.% at the bottom of all cores (Fig. 6). Consequently, the Fe/Al ratio, which eliminates effects from CaCO$_3$ and opal dilution and allows for the interpretation of Fe depletion or enrichment relative to detrital sources (Lyons et al., 2003), is stable in the upper approx. 400 cm at around 0.65-0.75 and increases to 1.2-1.5 at depth. The increase pointing to an Fe enrichment is much more pronounced in the westerly cores *Reference West* and *DEA West*, while the easterly cores show no substantial increase (*Small Crater*) or only
to around an Fe/Al ratio of 1.10 (*Reference East*). Mn/Al displays similar profiles, with higher ratios in *Reference West* and *DEA West* (0.3-1.3), while the other cores have similar ratios between 0.02 and 0.2 except for a few single layer outliers.

Manganese concentrations in the pore water increase with depth in varying gradients, asymptotically reaching maximum concentrations of 40-130 µmol/L at depths below 5-8 m (Fig. 7). Concentrations are lower in the
western areas and the *Small Crater* where nitrate does not get depleted (Fig. 3). Such a distinct difference between the sites can also be observed in dissolved Co concentrations. However, dissolved Co concentration profiles display elevated concentrations compared to bottom water already between 2 and 3 m, and show further increase below 6 m. The *Reference West* core exhibits the lowest Co concentrations. In contrast, dissolved Cu concentrations remain rather low and show no downcore trend.


**3.3 Redox-sensitive metals U, Mo, V, As and Cd: solid phase and pore water**

Dissolved concentrations of the redox sensitive elements U, Mo, and As as well as Cd are constant with depth in suboxic pore waters, and U and Mo also show straight profiles in the solid phase (Fig. 8). Arsenic and Cd could not be determined in the solid phase due to the formation of gaseous $AsF_5$ during HF digestion of the samples as

well as unreliable Cd measurements with the ICP-MS, respectively. Considerable peaks in the solid-phase and pore-water concentrations of U, Mo, and As (only pore water) are, however, visible for *Reference East* at depths 229.5 cm, 236.5 cm and 330 cm, where diffuse dark gray bands of approximately 1 cm thickness exist in the sediment (de Stigter, 2015). Vanadium concentrations peak at 229.5 cm in the solid phase (240 ppm) and the

concentration is still elevated at 236.5 cm (194 ppm), which is again reflected in the pore-water profiles. There is an additional peak in the solid-phase concentration at 290 cm, where the buried nodule was sampled, but no pore-water data exists for this exact layer. At *DEA Black Patch*, dissolved U, V, and Cu peaks coincide at 261 cm and U and Cd at 328 cm (Fig. 8). Solid-phase concentrations of U and V are also elevated in these layers (Fig. 8).

### 3.4 REY profiles and patterns: solid phase and pore water

Like Fe and P, REY concentrations increase with depth, especially at *Reference West* and *DEA West* (Fig. 6), and except for *Small Crater*. The sum of REY concentrations varies between approx. 180 ppm and 550 ppm (not shown). The buried nodules at *Reference West*, *DEA Trough*, and *Reference East* show similar to slightly lower REY concentrations than the sedimentary REY (see Nd in Fig. 6). Too little pore-water data is available to make statements about the concentration trend with depth. All solid-phase REY$_{SN}$ patterns show an enrichment of HREY

over LREY with La$_{SN}$/Yb$_{SN}$ ratios of 0.20-0.50, a negative Ce$_{SN}$ anomaly, and positive La$_{SN}$, Eu$_{SN}$, Gd$_{SN}$, and Y$_{SN}$ anomalies (Fig. 9). The negative Ce$_{SN}$ anomaly increases with depth (Ce$_{SN}$/Ce$_{SN}$*=0.6-0.3), the only exception being *Small Crater*, where the Ce$_{SN}$/Ce$_{SN}$* ratio remains at around 0.6 throughout the core. Y/Ho ratios range between 29 and 42, i.e. representing chondritic to super-chondritic values, and Eu/Eu* ratios are between 1.2 and 1.4. The Eu/Eu* ratios are, however, not pronounced enough to interpret a clear signal and are in the same range

as reported for seawater (Tostevin et al., 2016). REY$_{SN}$ patterns of the buried nodules show La$_{SN}$/Yb$_{SN}$ ratios of 0.40-0.44 similar to the sediment solid-phase REY, with negative Ce$_{SN}$ anomalies, slightly positive La$_{SN}$, Eu$_{SN}$, and Gd$_{SN}$ anomalies, and Y/Ho ratios of 27-30 (Fig. 9). Pore-water REY$_{SN}$ also show a HREY enrichment, a negative Ce$_{SN}$ anomaly and a positive Y$_{SN}$ anomaly (Fig. 10), similar to the sedimentary solid-phase REY$_{SN}$ patterns. All cores, except *Small Crater*, can be divided into an upper and a lower section based on the REY

concentration increase, increase in Fe/Al ratios, and a decrease of Ce$_{SN}$/Ce$_{SN}$* ratios: *Reference West* and *DEA West* at 4.5 m, *Reference South*, *DEA Black Patch* and *DEA Trough* at 6 m, and *Reference East* at 8 m (Fig. 9). The Fe/Al ratios remain steady in the *Small Crater* core, as well as the negative Ce$_{SN}$ anomaly. The first three above mentioned cores (*Reference West*, *DEA West*, *Reference South*) also have higher Y/Ho and La$_{SN}$/Pr$_{SN}$ ratios in their lower parts. The concentration increase is associated with the bottom of the green layer in cores *Reference*

*South*, *DEA Black Patch*, *DEA Trough*, and *Reference East*. In *Reference West* and *DEA West*, where no green layer exists, the concentration increase correlates with the color change from tan to dark brown at approx. 4.5 m and the increasing Fe and P concentrations at the corresponding depth. REY are most abundant, where a higher percentage of Fe(II) in the clay minerals prevails (*Reference West* and *DEA West*).

### 4 Discussion

**4.1 Paleoceanographic context: sedimentation history based on CaCO$_3$ and Ba preservation**

Sediments in the Peru Basin consist of clays and siliceous mud with some layers rich in CaCO$_3$ (Weber et al., 1995; Marchig et al., 2001) as depicted by the CaCO$_3$ and Ca concentration profiles of the GCs (Fig. 5). During times when the Carbonate Compensation Depth (CCD) deepened to depths below that of the seafloor, calcareous

skeletal material was preserved in the sediments upon burial. The present CCD is located approximately between 4200 and 4250 m water depth (Weber et al., 2000), slightly deeper than the water depths of the GCs presented here (4125-4208 m). Carbonate contents of more than 10 wt.% are present in the DISCOL area between 150-500 cm, concentrations and depths of $CaCO_3$ peaks vary slightly between the cores. Concentrations are lowest in the western cores *Reference West* and *DEA West*, which could be a sampling artefact due to sparse sampling, but both cores as well as *DEA Black Patch* have a second carbonate-rich layer at the base of the cores at approx. 800-1000 cm (Fig. 5). Carbonate dilutes other mineral phases, such as clay and Mn and Fe oxides, which is why concentrations of various (trace) elements in the solid phase, e.g., Al, Fe, Cu, Mn, Co, Ni, Zn, and REY are lower in carbonate-rich layers, while a few are enriched, e.g., Sr, due to their incorporation in the carbonate minerals.

The top of the carbonate-rich interval in the cores, located at approx. 150-200 cm, may tentatively be correlated to the 400 ka BP Mid-Brunhes event, when major carbonate dissolution occurred in the Pacific and after which carbonate was much less preserved in sediments (Weber et al., 1995; Weber and Pisias, 1999). The beginning of the upper $CaCO_3$-rich core interval at 500 cm may then potentially correspond to the onset of the deepening of the CCD 1.1 Ma ago, which continued until the Mid-Brunhes event 400 ka ago (Weber et al., 1995). The bottom carbonate layer is absent in some cores and based on our data set it is not possible to date it.

With 10-35 wt.% $CaCO_3$, the carbonate layers in our cores have similar concentrations as carbonate-rich layers reported previously for the DISCOL area (Weber et al., 1995, 2000). Weber et al. (2000) distinguished areas of higher bioproductivity and hence higher $CaCO_3$ input into the sediments in the northwestern and northeastern Peru Basin from less productive areas in the western and southern Peru Basin, including the DISCOL area.

Barium concentrations in marine sediments are often used as a marker for paleoproductivity but the use of this proxy depends on the reliability of the Ba record and that it was not subjected to alteration after burial of marine barite (Dymond et al., 1992; McManus et al., 1998; Gingele et al., 1999). In highly productive settings, authigenic barite formation can occur during diagenesis, while in most other settings under oxic and suboxic conditions, pore waters are saturated with respect to barite and solid phase barite is preserved (Reitz et al., 2004). Additionally, the biogenic barium concentration needs to be distinguished from the detrital barium concentration before it can be used as a paleoproductivity indicator (Gingele et al., 1999). We are therefore using Ba/Al ratios to only focus on biogenic Ba (Fig. 5).

Ba/Al ratios in the analyzed DISCOL sediments show elevated concentrations below approx. 350 to 450 cm, depending on the core, except for the core from *Small Crater*, which displays relatively constant concentrations throughout the core (Fig. 5). The layers with elevated Ba/Al ratios suggest a higher primary productivity and increased sedimentation rates at the time of deposition compared to sedimentation rates between 0.4 and 2.0 cm/ka reported previously for Peru Basin surface sediments (Haeckel et al., 2001). It is in these Ba enriched intervals that buried nodules were more commonly encountered, suggesting that increased sedimentation rates during times of higher productivity may have favoured nodule burial.

**4.2 Green layers**

Considering the small sampling area, the cores show a high heterogeneity of different layers and thickness of these layers. The color change from tan to green, visible in four cores (Fig. 3), represents the $NO_3^-$ penetration depth and the green color results from increased Fe(II) content in the nontronite, a process that has been well established for sediments in the Peru Basin (Lyle, 1983; Drodt et al., 1997; König et al., 1997, 1999). No dissolved Fe was detected in the pore water (limit of detection 0.5-1 µmol/L), confirming that there is no Fe-oxyhydroxide reduction taking place, mobilizing Fe into the pore water. Nitrate is present throughout the cores of *Reference West* and *Small Crater* (Fig. 3) and consequently, no green layers are observed, as Fe(III) dominates considerably in the nontronite. Nitrate is depleted at approx. 3 m depth at *DEA West* but no green layer is visible. Dissolved Mn concentrations are also lowest in these three cores (Fig. 7). This may be attributed to the lower POC contents of only 0.1-0.2 wt.% at depth compared to 0.2-0.4 wt.% that are found in the other cores without green layers (Fig. 4), which only allows for $NO_3^-$ and Mn(IV) reduction, but does not reach Fe(III) reduction in the electron acceptor sequence for POC degradation.

The cores with extensive green layers were located in depressions (*DEA Trough* and *Reference East*) and had few or no nodules on the seafloor (*DEA Black Patch*, *DEA Trough*, *Reference East*). Mewes et al. (2014) discovered that microbial respiration was higher at sites without nodules in the CCZ. This fits to the scenario in the Peru Basin, where fewer nodules occur in areas with more POC and therewith probably higher microbial activity. Most buried nodules, however, were found in depressions (Table 1) suggesting that their distribution and burial might be related to bathymetry-controlled sediment depocenters. Dissolving nodules and brown patches inside the green sediment layers (e.g., *DEA Black Patch*-497 cm and *DEA Trough*-585 cm) were found in the suboxic parts of the cores. The brown patches might be remnants of dissolving nodules because dissolving nodules impact their surrounding sediment, which is also visible in the 'halos' around the larger buried nodules. Fe(II)-rich sediment gets oxidized 'back' and is tan colored again (the 'halo'), as Fe(II) in nontronite is oxidized to Fe(III) (Russell et al., 1979; König et al., 1997; Dong et al., 2009), due to the provision of oxides by the nodules.

When clay minerals become concurrently enriched in Fe(III), they can transform into other clay minerals, such as glauconite or nontronite ( Pedro et al., 1978; Baldermann et al., 2015). Nontronite can form in three ways at the seafloor: (1) precipitation from hydrothermal fluids, (2) alteration of volcanic rocks, and (3) interaction of Fe (oxyhydr)oxides and biogenic silica at low temperature (Cole and Shaw, 1983). Hydrothermally derived nontronite has been found in Pliocene sediments of the Peru Basin and the adjacent Bauer Basin, but volcanic activity in the DISCOL area ended about 6 Ma ago (Marchig et al., 1999) and this age is not covered by the GCs presented here. Therefore, it is most likely that Fe (oxyhydr)oxides and (biogenic) silica form Fe(III)-Si complexes, which then develop into nontronite (pathway 3) ( Hein et al., 1979; Pedro et al., 1978; Cole and Shaw, 1983; Cole, 1985; Kashiwabara et al., 2018). This Fe(III) is provided by the buried nodules. The lack of high-temperature hydrothermal influence is also shown in the sedimentary $REY_{SN}$ patterns, which lack an $Eu_{SN}$ anomaly, a typical sign of high-temperature hydrothermally impacted sediments (Michard, 1989; German et al., 1990; Bau, 1991).

**4.3 Sedimentary Fe/Al**

Fe/Al ratios of 0.6-0.75 persist in the upper meters of all cores and throughout the core of the *Small Crater* (Fig. 6). This is in line with Fe/Al ratios of 0.6-0.7 of Pacific deep-sea sediments from other locations (Bischoff et al.,

1979; Paul et al., 2019). Elevated Fe/Al ratios of up to 1.3 or even above 3 in certain layers of our cores coincide with Fe/Al ratios of metalliferous layers in the central equatorial Pacific below approx. 5.5 or 8 m (Fe/Al: 1.3-1.7; Paul et al., 2019). Dissolving nodules analyzed in this study have Fe/Al ratios between 1.2 and 5.3, suggesting that the enrichment in the sediment could result from the dissolving nodules.

**4.4 REY as indicators for variability of deep-sea sediments**

The change in REY concentration with depth could be associated with past changes in sediment deposition – especially in cores *Reference West* and *DEA West*, where a color change from tan to dark brown is visible but no green layers. A second impact of REY concentration change might be related to a change in redox-zonation in cores *Reference South*, *DEA Black Patch*, *DEA Trough*, and *Reference East*, where the lower end of the green
layers coincides with the REY concentration increase. Small changes in the REY concentrations and SN patterns can be observed that correlate with other changes, e.g. changes in major element concentration (Fe, Al, P), or color (tan, dark brown, green). Small-scale variability is therefore also visible in the REY concentrations and SN patterns within the Peru Basin.

Correlations of REY and major elements help to elucidate phase associations of REY, which are important to understand before interpreting REY cycling. Neodymium (Nd) is used in the correlations to represent the REY. Correlations of solid phase Nd and major elements, such as Al, as indicator for detrital inputs, Mn as indicator for Mn oxides, Fe as indicator for Fe phases (Fe (oxyhydr)oxides or Fe-rich clay minerals), and P as indicator for phosphates – showed that Fe, Al, and P correlate positively with Nd (Figs. 11 and 12) while Mn shows no
correlation.

Iron-Nd correlations are positive in all cores (Fig. 11) and show the highest Pearson R coefficients of all, indicating the best fit for REY with Fe. At *Reference South* and *DEA West*, Fe also correlates with Al in the upper part of the cores (Fig. 11). The Fe-Al correlation points to the occurrence of an Fe-rich clay mineral. The carrier phase for the REY could therefore be a Fe-rich clay such as nontronite. Clay minerals have been postulated by others as the
primary phase controlling pore-water/solid-phase REY cycling ( Zhang et al., 2016; Abbott et al., 2019). The REY also correlate with Al at *Small Crater* and at *DEA West* until approx. 450 cm and at *Reference West* below approx. 450 cm, which matches the depth of the color change from tan to dark brown sediment in the latter two cores. It is unclear why only part of each core shows a correlation of Al with Nd and Fe and it is especially unclear why this
is sometimes the upper and sometimes the lower core section. Nevertheless, this finding corroborates the association of REY with Fe-rich clay minerals. Additionally, REY$_{SN}$ patterns of detrital clay minerals, such as illite or kaolinite, are flat due to their detrital origin (Cullers et al., 1975; Prudêncio et al., 1989; Tostevin et al., 2016) and, therefore, can be excluded here due to HREY enrichment and the pronounced negative Ce$_{SN}$ anomaly (Fig. 9). The sedimentary REY$_{SN}$ patterns with La/Yb $\ll$ 1, negative Ce$_{SN}$ anomaly, and positive La$_{SN}$, Gd$_{SN}$ and
Y$_{SN}$ anomalies are similar to REY$_{SN}$ patterns reported for nontronites (Fig. 9, Murnane and Clague, 1983; Alt, 1988; Mascarenhas-Pereira and Nath, 2010), which are expected to occur in these sediments because of the observed tan-green color change and the high Fe/Al ratio. The published nontronite REY$_{SN}$ patterns, however, refer exclusively to hydrothermally produced nontronites and the nontronite in cores from this study are not hydrothermally affected but rather derived from altered clay minerals or Fe (oxyhydr)oxides (e.g., Cole, 1985). To

the best of our knowledge, no REY data of nontronite that evolved from the combination of Fe (oxyhydr)oxides and biogenic silica exists that could be used for REY pattern comparison here.

Phosphorus correlates with Fe in cores from *Reference South*, *DEA West*, *Reference West*, and *DEA Black Patch*, which could be a sign of P bound to Fe phases. But P-Ca correlations in the Ca-poor parts of all cores, except *Reference East* are positive as well (Fig. 12), indicating a Ca phosphate phase. Ca-rich parts were excluded from this correlation since the high $CaCO_3$ contents obscure any P-Ca correlation. Since P and Nd also correlate in all cores, except *Small Crater* (Fig. 12), phosphates might play a role as a REY-controlling phase. The correlation of P and Nd in some cores is similar to results from large areas of the central equatorial Pacific, where REY are bound to (biogenic) Ca phosphates e.g., fish debris deposited in the sediments (Elderfield et al., 1981; Toyoda and Masuda, 1991; Toyoda et al., 1990; Toyoda and Tokonami, 1990; Kon et al., 2014; Deng et al., 2017; Kashiwabara et al., 2018; Liao et al., 2019; Paul et al., 2019;). There, Ca phosphates show middle REY (MREY) enriched patterns with no or negative $Ce_{SN}$ anomalies (Toyoda et al., 1990; Toyoda and Masuda, 1991; Paul et al., 2019). Apatite pellets with similar REY patterns as presented here (Fig. 9) were found on the Peru shelf (Piper et al., 1988), supporting the possibility of Ca phosphate control on REY in these sediments.

In conclusion, both, Ca phosphates and Fe-rich clays are potential REY-controlling phases based on the element correlations shown. Jarvis (1985) suggested a combination of Fe phases and phosphatic phases for the control of REY in Pacific metalliferous sediments. As Fe phases, but not Fe-(oxyhydr)oxides which we can exclude based on the $REY_{SN}$ patterns with no negative $Y_{SN}$ anomaly, release some REY to the pore water during recrystallization because the large ionic radii do not fit anymore in the smectite structure (Jarvis, 1985; Barrett and Jarvis, 1988), they are then available for scavenging by the Ca phosphate phase (Barrett and Jarvis, 1988; Kashiwabara et al., 2018). Clay minerals have similarly been described as a major phase influencing pore-water $REY_{SN}$ patterns during clay mineral dissolution and authigenesis (Abbott et al., 2019), which can then be scavenged by Ca phosphates (Zhang et al., 2016). Simple desorption from detrital clay minerals is unlikely the source determining the pore-water $REY_{SN}$ pattern, as the detrital clay minerals have no $Y_{SN}$ anomaly (Cullers et al., 1975; Prudêncio et al., 1989; Tostevin et al., 2016), but the pore water presented here has a positive $Y_{SN}$ anomaly. The matching pore-water and solid-phase $REY_{SN}$ patterns (compare Figs. 9 and 10) suggest that Fe-rich clay-phases release REY to the pore water or alter the pore-water REY pool during authigenic clay mineral formation, determining the pore-water $REY_{SN}$ pattern. The pore-water $REY_{SN}$ pattern  is then taken up by the Ca phosphates as they incorporate REY from the ambient pore water without major fractionation through coupled substitution, i.e. replacement of $Ca^{2+}$ by $REE^{3+}$ together with a monovalent element of similar size as Ca, e.g. $Na^+$ (Elderfield et al., 1981; Rønsbo, 1989; Jarvis et al., 1994). Similar $REY_{SN}$ patterns have been found in sediments in the DISCOL area and were explained to result from hydrothermal inputs and scavenging of REY from seawater (Marchig et al., 1999). Since hydrothermal inputs do not play a role in the sediments we investigate here, it is unlikely that hydrothermal activity affects the $REY_{SN}$ patterns in the GCs from this study. We propose that the incorporation of REY from ambient pore water is the dominant process resulting in the observed $REY_{SN}$ patterns. This is the same process as in the central equatorial Pacific (see e.g., Paul et al., 2019), but the pore-water $REY_{SN}$ pattern is different in the Peru Basin, leading to different patterns in the solid phase.

Even though the same incorporation process into the solid phase takes place in the Peru Basin and the CCZ – two Pacific nodule areas in the focus of investigating mining-related disturbances – the solid-phase $REY_{SN}$ patterns are different due to the different pore-water $REY_{SN}$ patterns. While the same general pattern (HREY enrichment, negative $Ce_{SN}$ anomaly, positive $Y_{SN}$ anomaly) is observed in all cores in the Peru Basin, they differ from the $REY_{SN}$ pattern observed in the CCZ (MREY enrichment, no or negative $Ce_{SN}$ anomaly). The REY are therefore a suitable parameter for the interregional comparison of sediments.

**4.5 Dissolved Mn, Co, and Cu**

Dissolved Mn concentrations increase with depth and from west to east (except for *Small Crater*), thus mirroring the solid-phase Mn in these cores, including the surface sediments, where concentrations are higher in the west than in the east (Paul et al., 2018). Similarly, dissolved Co concentrations at depth are higher in the east than in the west and vice versa in the solid phase except for *Small Crater* (Fig. 6). Both western cores and *Small Crater* have the lowest POC concentrations and the deepest $NO_3^-$ penetrations depths (Fig. 3). Manganese oxides are therefore less utilized as electron acceptors during the degradation of organic matter in these cores and less Mn is released to the pore water.

The marked increase of dissolved Mn and Co concentrations at depth might also be related to the release of trace metals from buried, dissolving nodules. *Reference South*, *DEA Black Patch*, *DEA Trough*, and *Reference East* show highest dissolved Mn and Co concentrations at depth and show green layers, in which nodules are dissolving.

Copper does not display the west-to-east-trend in the pore-water profiles and does also not show an increase at depths where Mn and Co are enriched in the suboxic zone. A deviation of Cu from the behavior of Mn, Co, Ni etc. has already been found in our previous study (Paul et al., 2018). While Mn, Co, and Ni are largely controlled by Mn oxides and their reduction during POC degradation (Klinkhammer, 1980; Heggie and Lewis, 1984; Shaw et al., 1990), Cu is largely controlled by the release from organic matter during early diagenesis and only partially due to association with Mn oxides (Klinkhammer, 1980; Shaw et al., 1990).

**4.6 Redox-sensitive metals Mo, U, As, and V: solid phase and pore water**

The redox-sensitive metals Mo, U, As, and V are soluble under oxic conditions and are bound to the solid phase under anoxic conditions in the sediment (Elbaz-Poulichet et al., 1997; Beck et al., 2008; Wang, 2012). They display conservative type profiles in oxic pore waters and are all associated with cycling of organic material, Mn (for Mo, As, V), and Fe (for U, As) (Beck et al., 2008; Telfeyan et al., 2017). In the suboxic sediments presented here, profiles are largely conservative (Fig. 8), except few peaks, and in the same range as concentrations in oxic pore waters in the Peru Basin (Paul et al., 2018). Therefore, conditions in the Peru Basin sediments are likely insufficiently reducing to lead to a redox change for these elements with depth. An exception are the gray bands in *Reference East*, where U, Mo, V, and As concentrations peak in the solid phase and pore water, dissolved Co concentrations are low (even below the LOQ at an average of 0.14 µg/kg) and dissolved Mn concentrations are slightly lower than in the surrounding sediment above and below (Fig. 7). This might be a sign of locally oxic conditions releasing U, Mo, As, V, and Cd into the pore water but removing Co and Mn. Elevated concentrations of U, Mo, V, and As in the pore water are also possible due to the chemical equilibrium between the high concentrations in the solid phase and the pore water, so that oxic conditions might not necessarily be required, but

the controlling process cannot be identified with certainty. Total dissolved S in the pore water is not elevated in these layers, while at 238 cm, where another gray band was sampled for solid-phase S analyses, elevated concentrations of 0.54 wt.% S were measured compared to ~0.3-0.4 wt.% S in most of the core, possibly a sign of anoxic-sulfidic deposition of material or the presence of barite, but this cannot be said with certainty.

The *Reference East core*, as well as the *DEA Black Patch* core, are located in areas with few or no nodules at the seafloor surface. In addition, *Reference East* is located at greater water depth (56-91 m deeper than the other sites). Deposition of different material – also more organic material that might lead to periods of anoxic conditions – is the standard explanation for enrichments of U, Mo, As, and V in other settings, but the observations here can most

likely not be explained by anoxic conditions because of low POC contents (~0.3-0.5 wt.% in the *Reference East* core). The occurrence of these gray bands with elevated U, Mo, As, and V concentrations is striking but we cannot clearly explain their source.

The solid-phase and dissolved U, V, and Cu concentration peaks in *DEA Black Patch* suggest the presence of a

Cu-rich uranium-vanadium phase. This is known from oxidation fronts in turbidites in North Atlantic clays, where U, V, and Cu are enriched in the solid phase (Colley et al., 1984; Colley and Thomson, 1985). The metals are mobilized during organic oxidation of the turbidite material, migrate downwards, and are immobilized at depth (Colley et al., 1984). They are preserved by burial of other material on top (Colley and Thomson, 1985). In the Peru Basin, solid-phase peaks of Cd, Cu, and V have been attributed to the downward progression of the

oxic/suboxic boundary during glacial/interglacial cycles which is slowed down by the reactive Fe(II) layer in the clay minerals, and where this oxic front reaches the reactive Fe(II) layer, heavy metals such as V and Cu can be precipitated (authigenic precipitation of U, V, and Cu) (König et al., 2001; Koschinsky, 2001). A similar process during organic oxidation might have taken place at *Reference East*.

## 5 Conclusions

The analyses of seven GCs from the DISCOL area show that a deep-sea basin can be highly heterogeneous even on small spatial scales. The variability is visible in organic matter content (POC) and related differences in $NO_3^-$, Mn, Fe (and REY) concentrations as well as for individual layers where redox sensitive elements such as U, Mo, V, and As are enriched. Especially *Small Crater* is different in the measured parameters from the other cores: no green layer and generally more layers with dark brown sediment, Fe/Al ratios remain constant, and REY only

correlate with Fe and Al throughout the cored sediment. Since these exceptions correspond to special locations, such as lower lying areas without or with less nodules where redox sensitive metals are enriched and the *Small Crater* where a different deposition environment might prevail, the importance of small topographical changes is presented as a possible explanation for the geochemical variations. The importance of small variations in depositional environments has been underestimated in the deep sea and this study showed, how extensive the

effects of the depositional area can be on the various geochemical parameters presented here. Variability, however, could be higher at DISCOL than in areas further away from continents, because the DISCOL area might be more impacted by continental inputs and higher primary productivity than e.g., the CCZ, as the DISCOL area is located at the southern edge of the equatorial high productivity zone.

The results call for caution when extrapolating findings from a small set of samples to larger ocean areas. With respect to deep-sea mining, the results show how variable the deep-sea floor can be and that extensive baseline studies are necessary before the onset of mining and impact analyses. This has been stressed by various advocates for the preservation of the deep-sea ecosystem (Glover and Smith, 2003; Mengerink et al., 2014; Van Dover et al.,

2014; Schindler and Hilborn, 2015). Since the geochemical composition of the sediment, including POC content and redox conditions, has a major impact on microbial processes in the sediment and associated biological life, this small-scale heterogeneity may also be relevant for biological productivity and diversity in the deep sea, as well as biological recovery after deep-sea mining disturbances.

Another interesting finding of this study is the influence of dissolving nodules on the surrounding sediment and geochemical cycling, e.g., in the form of visible 'halos' in the sediment or increased Fe/Al ratios and dissolved Mn and Co concentrations in the pore water. These dissolving nodules can therefore lead to significant small-scale differences in the mineralogical and chemical composition of sediment cores and care should be taken that such signatures are not misinterpreted as e.g., hydrothermal influence.

**Author contribution**

SP, MH, AK: research design. SP: sampling, trace metal analyses with contributions from RB. MH: sampling, POC, $CaCO_3$, and nitrate data collection. SP: data interpretation with contributions from MH, MB, and AK. SP prepared the manuscript with contributions from all co-authors.

**Competing interest**

The authors declare that they have no conflict of interest.

**Acknowledgements**

Thanks to the crew of RV SONNE and chief scientist Jens Greinert on cruise SO242/1, who enabled our sampling. We thank especially Henko de Stigter for the valuable core description made during SO242/1 and edits to the manuscript. Our great appreciation goes to Katja Schmidt, Annika Moje, Inken Preuss, Tim Jesper Suhrhoff and

Laura Ulrich for help with sampling and laboratory analyses in the geochemistry lab at Jacobs University Bremen. We are indebted to Meike Dibbern, Bettina Domeyer, Anke Bleyer, and Regina Surberg for their analytical support during the RV SONNE cruise and at GEOMAR. Thanks also go to Anne Hennke and Jens Greinert, GEOMAR, for providing the original bathymetry map, Laura Haffert from GEOMAR for providing the depth correction of the GCs, and Charlotte Kleint, Jacobs University Bremen, for helpful comments during the writing process. The

work was funded by the German Federal Ministry of Education and Research in the framework of the JPI Oceans project MiningImpact (grant no. 03F0707A+G).

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

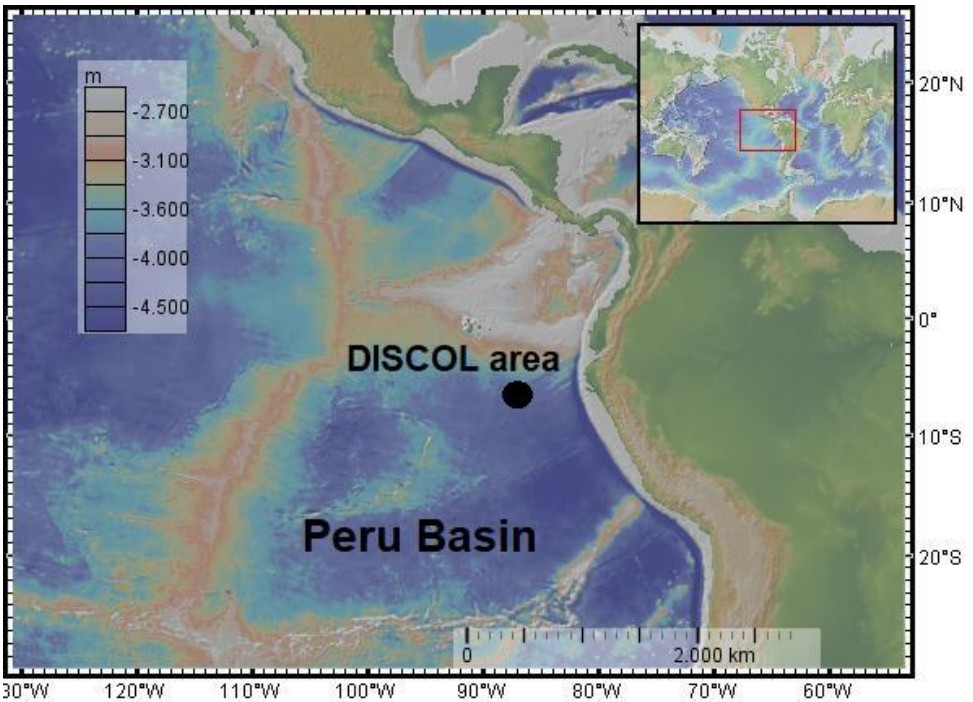

**Figure 1: The Peru Basin with location of the DISCOL area. The map was created using GeoMapApp**
15 **([www.geomapapp.org](http://www.geomapapp.org)), CC BY, and its integrated default basemap Global Multi-Resolutional Topography (GMRT),**
**CC BY (Ryan et al., 2009).**

**Table 1: Overview of sampled cores.**

| Sample ID SO242/1 | Area | Location | Water depth [m] | Core length [cm] | No. of samples | Nodule on top | Buried nodules |
|---|---|---|---|---|---|---|---|
| 38GC1 | Reference South | 7°07.537' S 88°27.047' W | 4161 | 917 | 13 | yes | no |
| 51GC2 | DEA West | 7°04.411' S 88°27.836' W | 4148 | 978 | 16 | no | no |
| 84GC3 | DEA Black Patch | 7°03.951' S 88°27.093' W | 4146 | 947 | 17 | no | no |
| 89GC4 | Reference West | 7°04.562' S 88°31.577' W | 4125 | 958 | 11 | yes | 1 |

| 100GC5 | DEA Trough | 7°04.342' S 88°27.442' W | 4151 | 878 | 14 | no | 3 |
| 123GC6 | Reference East | 7°06.045' S 88°24.848' W | 4208 | 921 | 16 | no | 4 |
| 132GC7 | Small Crater | 7°03.369' S 88°26.031' W | 4152 | 936 | 12 | no | 2 |

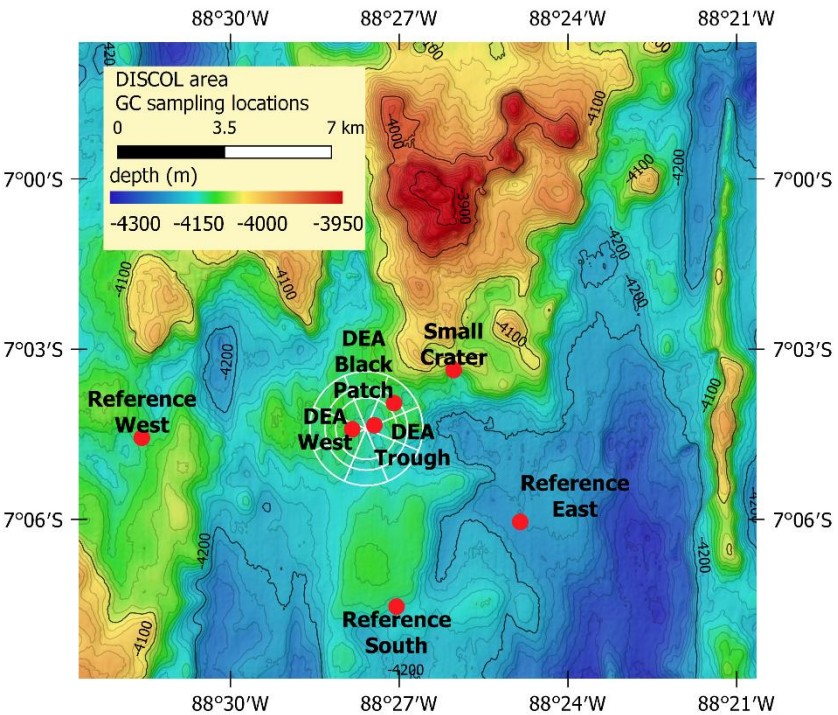

**Figure 2: GC sampling locations in the Peru Basin. The circle indicates the DISCOL experimental area (DEA) that was traversed with a plow harrow. Created with QGIS with bathymetry data provided by Anne Hennke and Jens Greinert, DSM group, GEOMAR.**

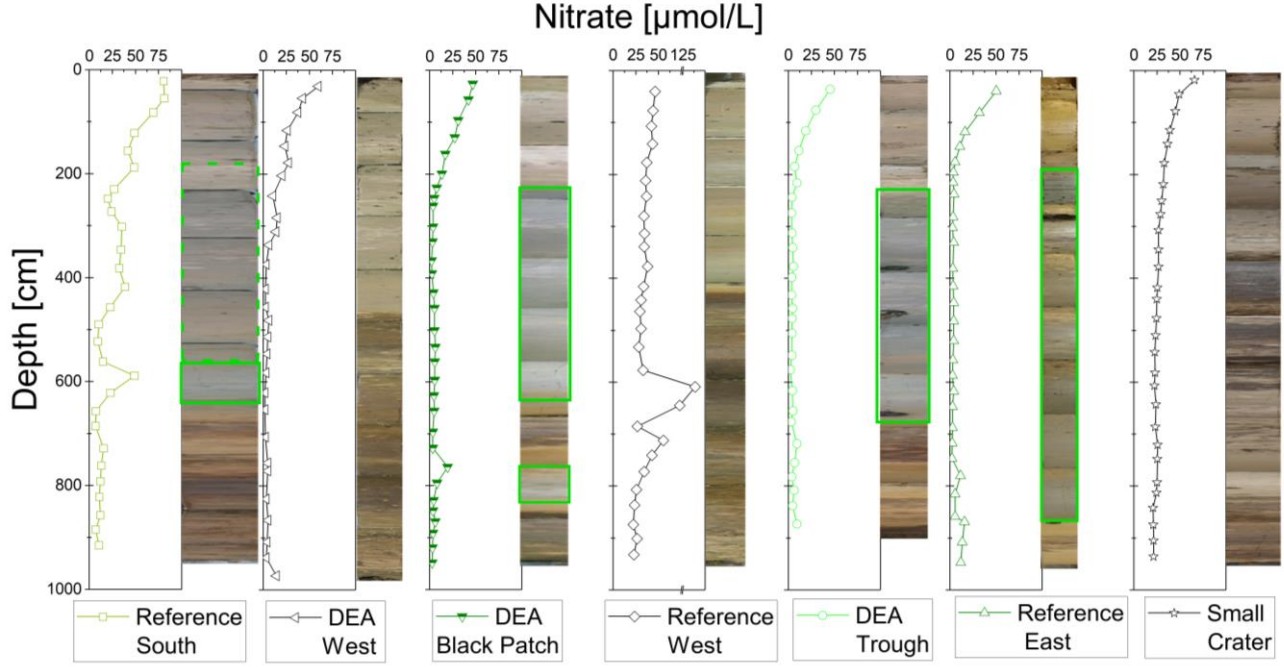

**Figure 3: Combined photos of the individual GCs with corresponding nitrate profiles. Green layers are marked with green boxes.**

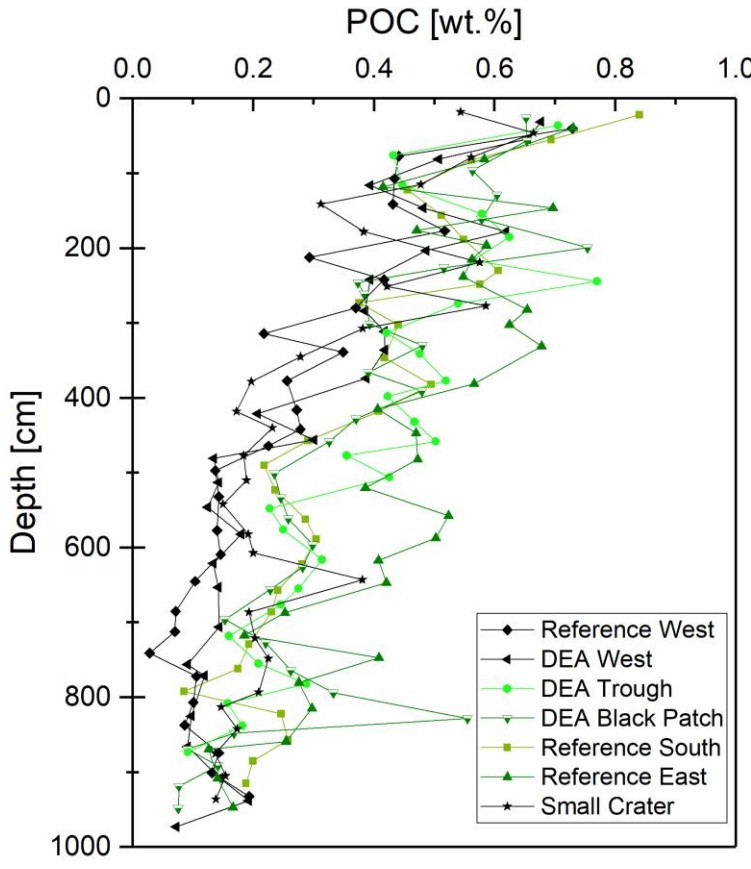

**Figure 4: POC profiles of the GCs.**

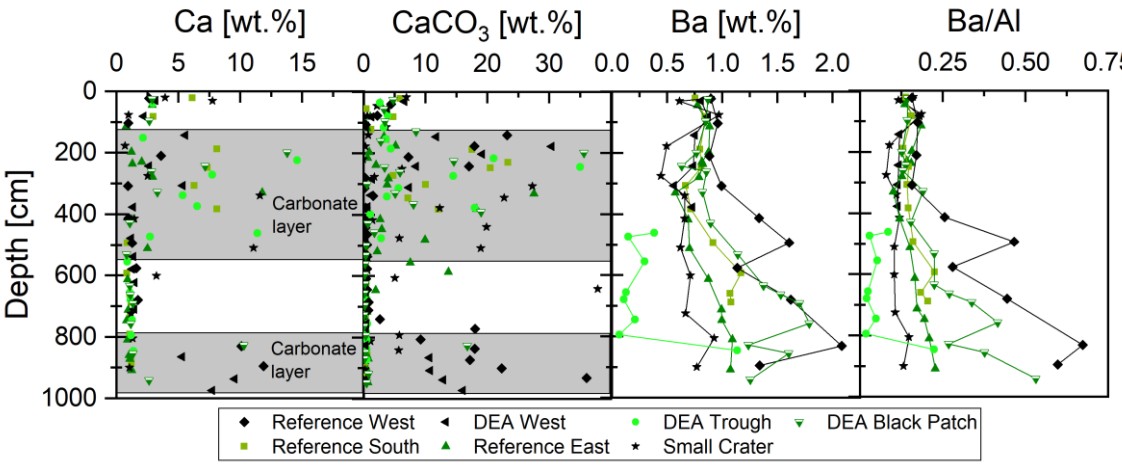

**Figure 5: Depth profiles of solid-phase Ca, CaCO₃, and Ba concentrations, as well as Ba/Al ratios. Core intervals with higher contents of preserved carbonate are shaded in gray.**

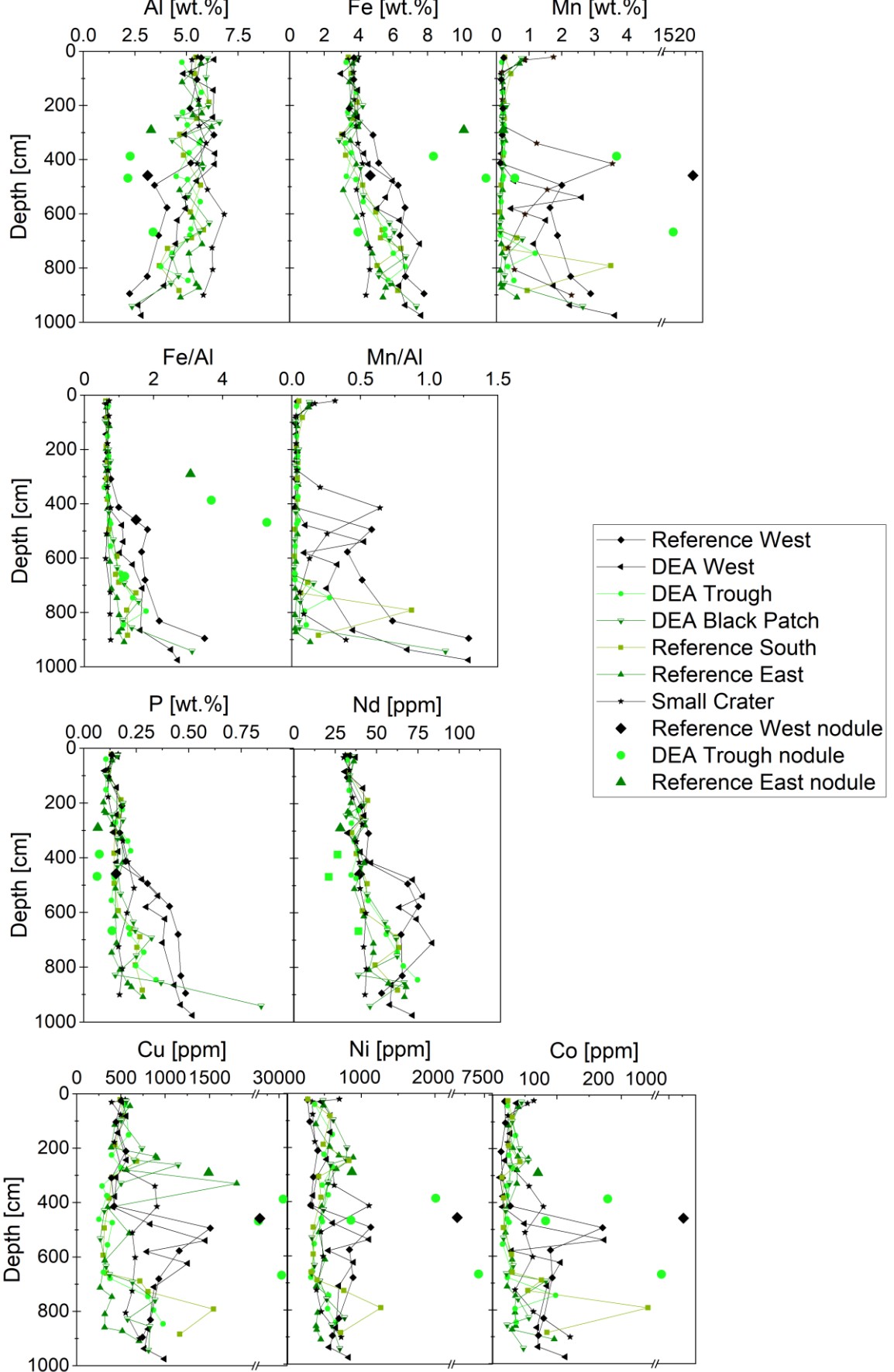

**Figure 6: Solid-phase Al, Fe, Mn, P, Nd, Cu, Ni, and Co concentrations in the sediment cores including those of the buried nodules at Reference West at 458 cm, at DEA Trough at 387 cm, 468 cm and 667 cm, and at Reference East at 290 cm depth. Nd is shown as a representative of the REY. Fe/Al and Mn/Al ratios (for the latter no data for the nodules is shown) are also displayed as depth profiles, focusing on the Fe and Mn enrichment in relation to continental sources (Al).**

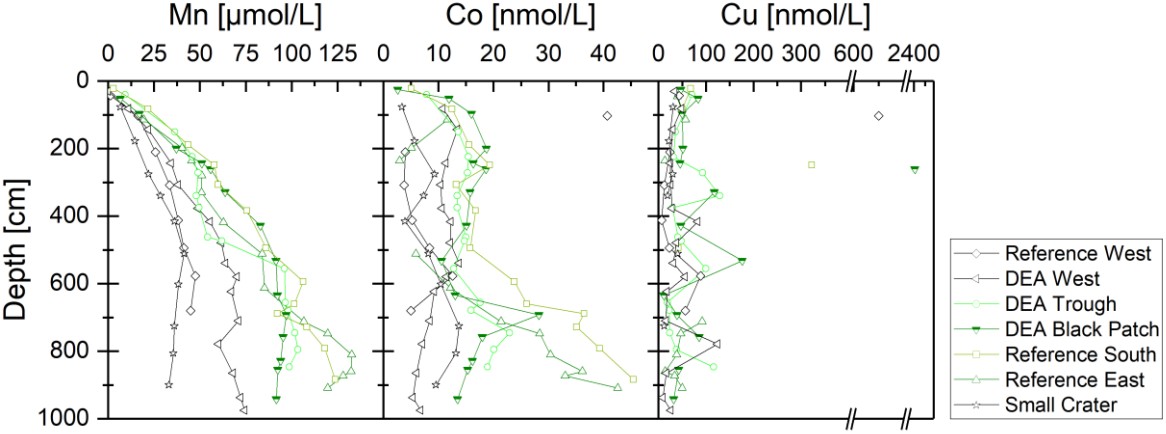

**Figure 7: Dissolved Mn, Co, and Cu concentrations in the pore water of the sediment cores. No pore water could be extracted from buried nodules.**

**Figure 8: Top: Solid-phase concentrations of U, Mo, and V. Concentration peaks are visible at 229.5, 236.5 cm and 330 cm for Reference East coinciding with the gray bands in the sediment (see pictures on the right). In this core, also a dissolving nodule was found at 290 cm (see pictures on the right). Bottom: Dissolved concentrations of U, Mo, V, As, and Cd in the pore water. Depths 229.5 cm and 290 cm of Reference East were not measured. Concentration peaks are visible at 236.5 cm and 330 cm for Reference East coinciding with the gray bands in the sediment (see pictures on the right).**

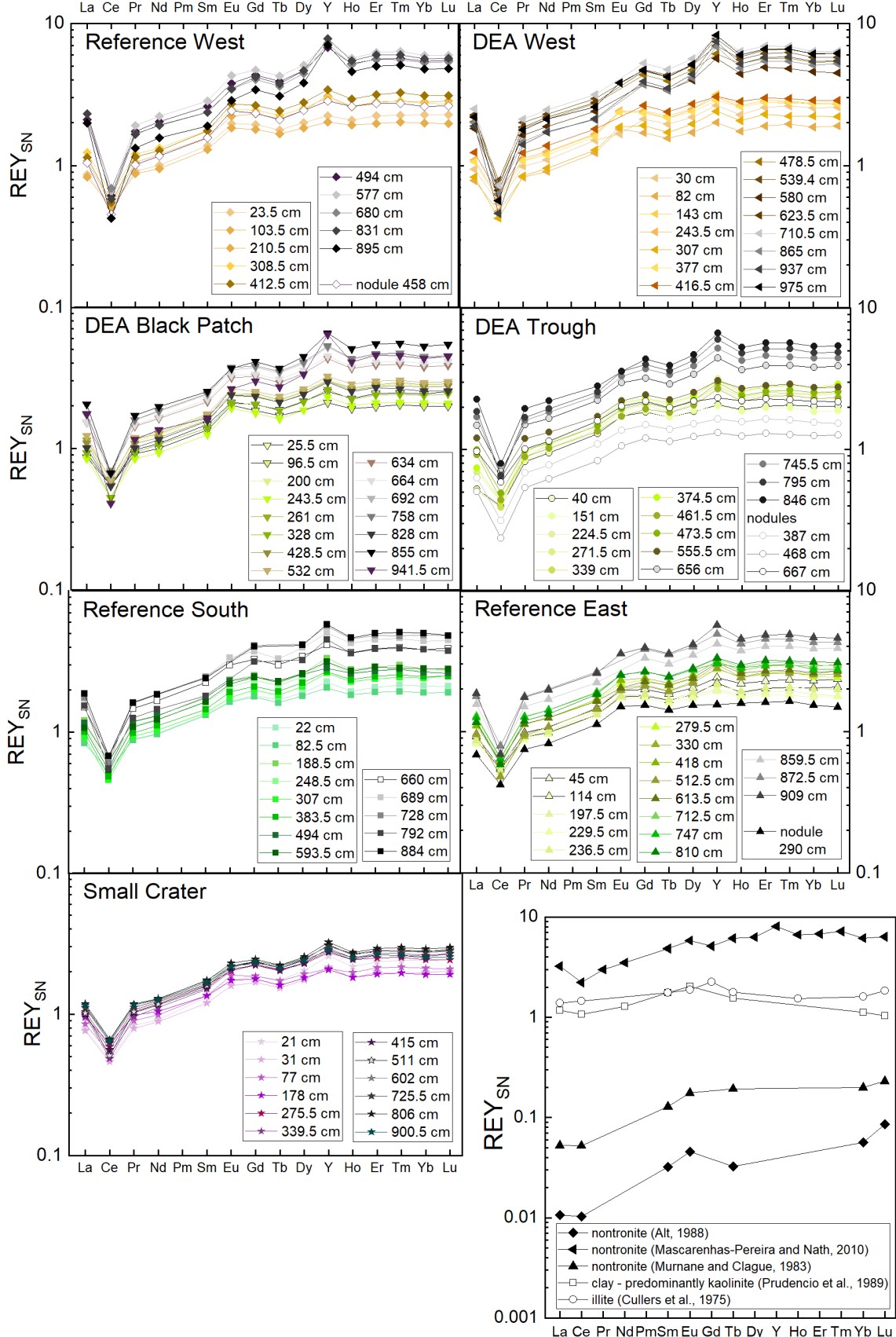

**Figure 9:** REY$_{SN}$ patterns of the seven cores from this study and for the clay minerals nontronite, illite, and kaolinite from literature for comparison.

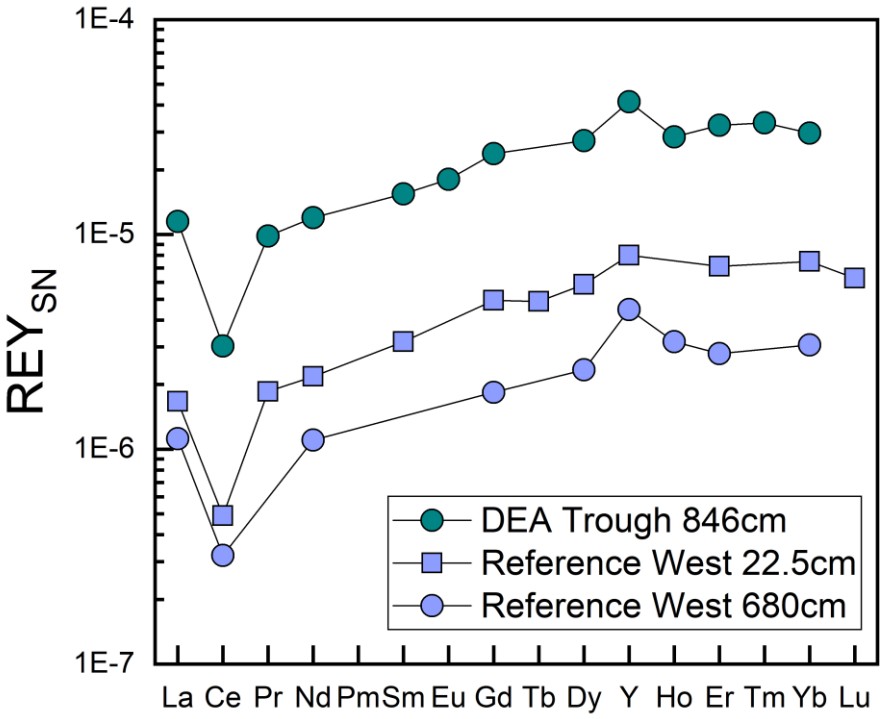

**Figure 10: Measurable pore-water REY$_{SN}$ patterns from the Peru Basin.**

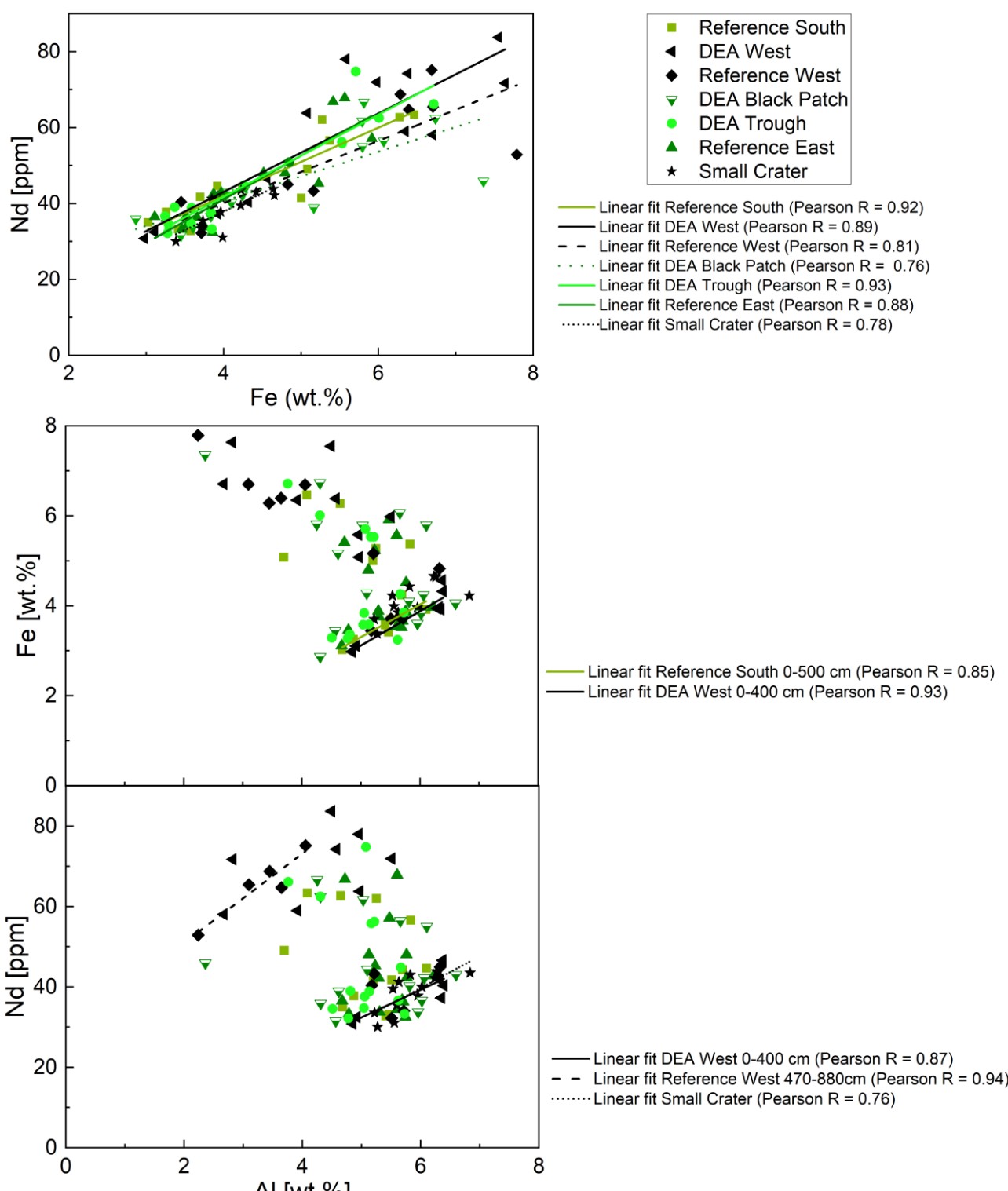

**Figure 11: Top: Fe-Nd plot and correlations for all cores. Pearson R coefficients show positive correlations of REY with Fe for all cores. Middle: Al-Fe plot. Only positive correlations for the upper parts of Reference South and DEA West are shown. Bottom: Al-Nd plot. Only positive correlations for the upper part of Reference South, as well as for the lower part of Reference West and the entire Small Crater core.**

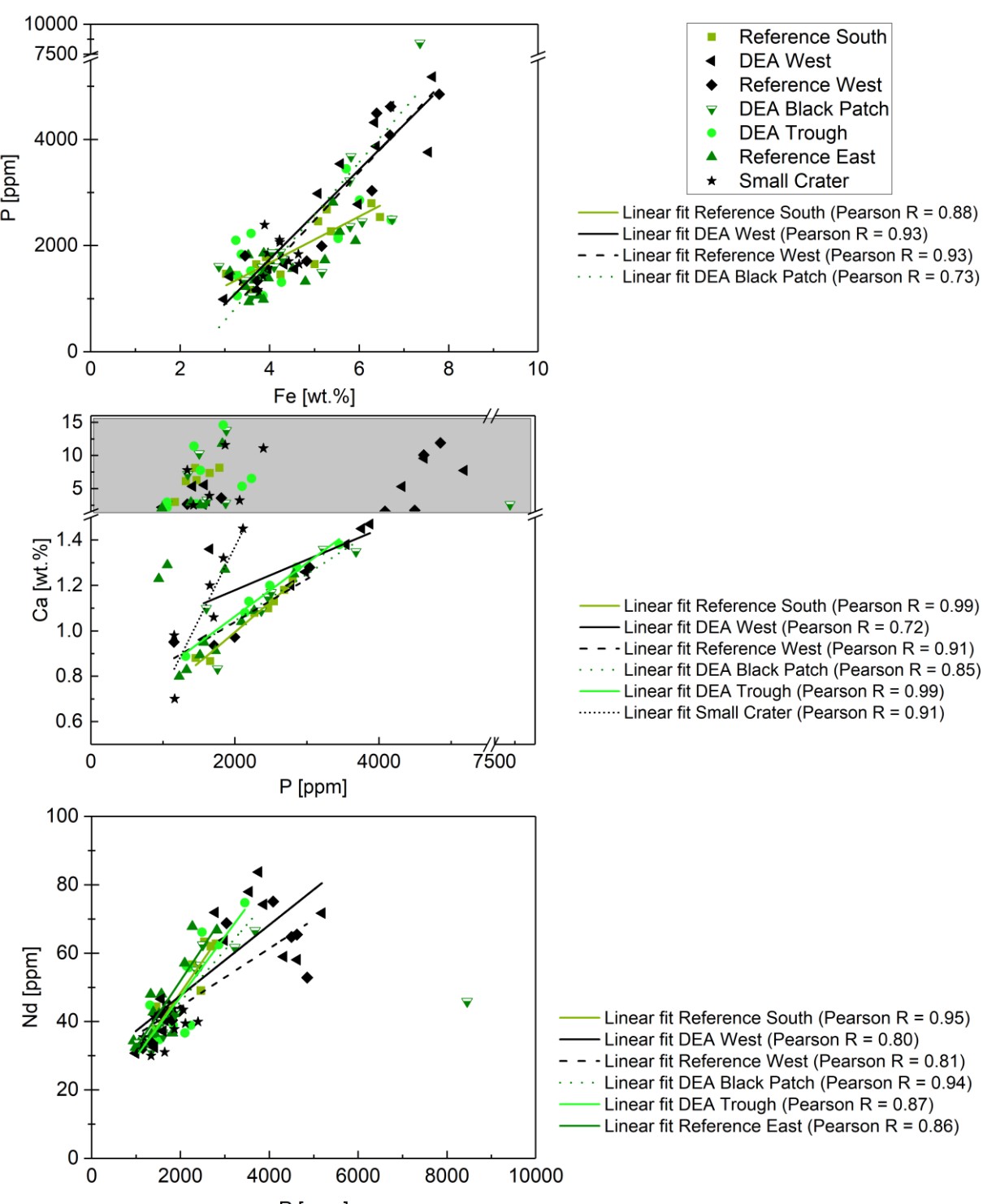

**Figure 12: Top: P-Fe correlations for Reference South, DEA West, Reference West, and DEA Black Patch. Middle: P-Ca correlations for samples with Ca concentrations below 1.5 wt.% except for Reference East where P and Ca do not correlate. Samples with Ca concentrations above 1.5 wt.% were excluded from the regression analyses because most of the Ca is then not bound in Ca phosphates. Bottom: P-Nd correlations for all samples except Small Crater where P and Nd do not correlate and excluding the DEA Black Patch sample with exceptionally high P concentrations.**