# Peer review of "Small-scale heterogeneity of trace metals including REY in deep-sea sediments and pore waters of the Peru Basin, SE equatorial Pacific"

_Biogeosciences, 2019_

## Referee Comment (RC1) · Anonymous Referee #1 · 12 Sep 2019

In the manuscript bg-2019-274 "Small-scale heterogeneity of trace metals including REY in deep-sea sediments and pore waters of the Peru Basin, SE Equatorial Pacific," Paul and coauthors demonstrate that 1) the depositional environment of a specific coring location matters even within a small geographic range and that 2) iron rich clays and phosphates likely control the REY of the sediments.

The authors use the variability of redox sensitive metals, organic matter, and REY between several distinct depositional environments in the Peru Basin to demonstrate that a given core may not be representative of a bigger area. While definitely true, I feel

the authors over-sell a broad 'small-scale heterogeneity' emphasis while underplaying the fact that the sites were deliberately chosen in different depositional environments (including the title). Nuanced, but this data set clearly shows the importance of considering depositional environment when selecting sites and the authors don't really articulate that point, instead the message comes across as a more dismissive 'they're all different anyways.' In presenting this data set, the authors also use solid and fluid phase measurements to address the sediment phases controlling the REY of the sediments. The authors argue that Fe-rich clay mineral and phosphates are the controlling phases based on Fe/Nd, Fe/Al, Fe/P, and P/Ca correlations. I believe the authors are likely correct on this assumption, but am confused as to the lack of recent citations in support of this conclusion.

Generally, the manuscript is well written and adequately cited, however as alluded to above the manuscript seems weighted towards older (pre 2000) citations despite many of these topics growing rapidly over the last decade. The only aspect that was a bit challenging to follow was the jumps between solid phase and fluid phase, mainly as the approach to this wasn't consistent (e.g. 3.2 and 3.3 split up solid and fluid phases, then 3.4 was solid and fluid) and there were a couple of key details missing from the methods section. A few more specific comments are included below, but overall I feel the authors present a great data set that is relevant and appropriate for publication in Biogeoscience with minor revisions.

Specific comments: P1

31- "only a small part" – has this been quantified? i.e. more impactful if you can at least assign an order of magnitude to it (0.1%? 10%?)

35-36 the number of pore water studies in the last decade has drastically increased but citation list heavily weighted towards earlier ones (e.g. Deng et al 2017 Scientific Reports; Kim et al 2012 Chemical Geology; Schacht et al. 2010 Journal of Geochemical Exploration; Abbott et al 2019 Frontiers in Marine Science)

[Figure]

P3

20-21 again seems like a gap in including recent literature other than the author's own 2019 paper

30 ".. their detrital orgin." reference?

In terms of REE patterns/Ce anomalies: Kang et al. 2014 Journal of Asian Earth Sciences; Kon et al. 2014, Resource Geology

P4

section 2.1 unclear from the next the number of gravity cores taken and their depositional environments (with the exception of the description of the volcanic crater)

25-32 Were the cores sampled in ambient air? How long did the split cores sit before analyses? How are concerns about oxidising pore fluid before centrifugation addressed if sampled in ambient air?

30 Filter size is important – many of the other work compared to is 0.45 micron – not necessarily a problem, just needs to be explicitly recognised

P5

1 Incomplete digestions not mentioned again- need to be discussed in terms of implications

4 1-2 sentences explaining what the apex Q does would help your reader rather than having to go to the ESI website

P8

section 3.5 this section is far less detailed than the precedent set in the earlier sections, stay consistent. Variability? Trends?

P9

section 4.1 one more sentence here reminding readers of the depth of the sites 4125 and 4216 when discussing modern CCD would be helpful

P10

section 4.2 again seems lack recent references (e.g. Baldermann et al 2015 Nature Geoscience; Dong et al 2009 American Mineralogist; Huggett et al 2017 Clay Minerals)

P11

section 4.4 discussion should frame results in context of similar observations in the literature— e.g. lots of work showing Nd and Fe relationship

28-30- "Even though it is. . . Fe-rich clay minerals." this sentence is a bit awkward to read but makes one of the main points, worth reworking into 1-3 clear sentences instead of a complicated one.

34- 'reported for nontronites." By who? reference?

P12

11, 27-28 This jumps from the last thought- where is the pore water getting the REY and how do they compare to the sediments? Needs a few more sentences walking the reader through the logic.

P13

sections 4.5-4.6 hard to track dissolved versus solid- separate clearly or consistently label

11 "They display. . . in oxic waters" is this a general statement or also observed at these stations

28 is the mineralogy of the grey bands known?

P14

**BGD**

7-10 This is a very fair description of the spatial heterogeneity in terms of the different depositional environments

Figures/Tables

Fig 1) the red text is hard to read

Table 1) why is 84GC3 different significant figures for water depth then the rest of the sites?

Figure 2) Can the color bar be larger- very hard to read the text in the key

Figure 9) color coding on this figure is great- makes it really easy to follow what is going on!

Figure 11) is this the same key as figure 12?

---

## Referee Comment (RC2) · Anonymous Referee #2 · 12 Sep 2019

This study addresses the biogeochemical heterogeneity of deep sea sediments, which are often under-sampled but of great importance to our understanding of the global ocean system. The authors present pore water and solid phase trace metal, REY, carbon, nitrogen, and phosphorus data from six sediment cores in the Peru Basin and observe differences in sediment composition that may be related to variation in organic carbon contents, bottom topography, or sediment source. This is an impressive data set that will help expand and deepen our understanding of Pacific deep sea sediments. I fully agree with the authors' conclusion that caution should be exercised in extrapolating findings from a few cores to cover broader areas of the deep sea, and find this to be a valuable scientific contribution in itself. However, I do have some broad concerns about the use and presentation of the data, consideration of confounding variables, and the general frame of the paper. My largest concern is the fact that half the sites investigated in this study were subject to a disturbance and recolonization experiment thirty years ago, while the other half are pristine. In my opinion, there is not enough consideration of this potentially confounding variable, and how the impacts of ploughing could have caused some of the observed heterogeneity in the sediment. This is not to say the DISCOL experiment invalidates the results of this study; in fact, I think that a greater focus on the differences (or lack thereof) between the DEA and undisturbed sites would make a much more compelling frame for the paper. Additionally, the paper would be greatly strengthened by a more thorough discussion of how the results of this study, as a long term follow-up to the DISCOL experiment, relates to deep sea nodule mining and could inform future mining decisions. On the other hand, if the authors feel this study does not have a strong connection to current mining activities and decisions, then this should not be mentioned (e.g. Page 2, Line 12 and Page 14, Line 16), as the connection may mislead readers. Finally (in full acknowledgement that I am not an expert in rare earth elements), after reading the paper I was left uncertain about the usefulness and relevance of the REY data set in the frame of the study. It was unclear to me what further information the REY data imparted regarding biogeochemical processes and variation between the sites that was not apparent in the other (trace metal, carbon, etc.) data sets. This aspect of the paper could be improved by more background on REY in the introduction and a more detailed discussion of interpretation of the REY results in the frame of biogeochemical differences between the sites and/or the impact of polymetallic nodules at the surface or buried in the sediment. Overall, I support this paper for eventual publication after revisions in the areas described above. Below are some line edits and specific comments to guide revision of the manuscript.

(Note: P2. L20 means page 1, line number 20. For future reference, it would be helpful to have continuous line numbering in the manuscript, rather than line numbers that

restart every page.)

Abstract P1. L15 and L23: Be careful to clarify whether the heterogeneity referred to is between sites or between depths at a single site. The "variability" in line 23 seems as though it is referring to Mn and Co concentration peaks with depth, rather than differences between the sites.

Introduction

P1. L34: What is meant by biogeochemical heterogeneity, exactly? Different processes? Different carbon contents? Simply giving a few examples of relevant biogeochemical parameters that vary between sites would be helpful.

P1. L34-35 "In the past, few spread-out samples were collected for pore-water and solid-phase geochemical analyses" As written, the sentence does not emphasize the sparse nature of past sampling. Rephrase to something like: "In the past, cores collected for pore water and solid phase geochemical analyses have been sparse and separated by large distances." I'm sure there's a better way to word that, but hopefully you understand what I mean.

P1. L36: "on small spatial scale" revise to "on small spatial scales."

P2. L1: "could show" revise to "showed"

P2. L2: "studies of few samples" revise to "studies of a few isolated samples" or something similar.

P2. L12: How does the heterogeneity discussed in the paper relate to deep-sea mining? Will the results help inform mining decisions? Do they imply that mining does not have a significant impact on sediment biogeochemistry? If there is not a strong connection between the results and mining, I would minimize discussion of mining except to explain the reason for the DISCOL experiment.

P2. L23: "Mineralogical investigations of long cores were conducted extensively" This

seems to contradict the previous sentence.

Consider placing Section 1.3 before Section 1.2, so that the reader gets an idea of the study area before learning about previous work in the area. Learning about the sediment biogeochemistry and the presence of nodules will help the reader understand why the mining experiment occurred here.

P2. L31-37: Throughout the paper, the authors rely on sediment color to make assumptions regarding the geochemical composition of the sediment. Color can be a useful indicator, but should be backed up by true geochemical data. If such data exists, please include it in this paragraph (and others discussed below). If it does not, make this clear to the reader and be transparent that some of your mineralogical assumptions are based solely on color and may not be entirely reliable. For example, in line 35: "color change typically indicates re-oxidation" or in line 34 "The Fe(III) to Fe(II) redox boundary is assumed to occur where the sediment color changes from tan to green."

P3. L10-12: Here, I'm not certain that the sediment colors provide any useful information, since they should not be solely relied upon to determine geochemical composition later in the paper.

P3. L18: Fractionation associated with which processes? Again, I am not a rare earths expert, so it would help me to understand what processes REY fractionation can indicate.

General: - I would like to have more background on nodules— what are they composed of, how are they formed and how do they relate to the biogeochemistry of the sediment? How do the nodules "dissolve" and form the observed haloes in the sediment? - There should be consideration of the relationship of topography to sediment heterogeneity. It seems intuitive that the sediments will be heterogeneous if the topography is as varied as it is, and this is mentioned in the Conclusions but should be included in the Introduction and Discussion as well. - I am curious whether the sediments within the DISCOL area have the same redox zonation? Parts of the zonation must have been removed, but have they re-established since 1989? Discussion of this would help the reader understand the similarities and differences between the DISCOL sites and the undisturbed sites. - If the authors decide to maintain deep sea mining as a part of the "implications" of this study, there should be more background in the Introduction on mining in the area—what is mined, and how?

Methods

P4. L12: The "Therefore" is unnecessary. In fact, this sentence should go after the description of the disturbance experiment, maybe at the end of the paragraph.

P4. L20-22: I am not convinced that the ploughing had no effect on the sediment, or that the loss of sediment during coring removes that effect. The 20 cm lost from the ploughing was removed 25 years ago; the 20 cm lost in GC sampling was lost the instant the core was taken. Also, shouldn't the GC cores in the disturbed sites also lose 20 cm, so overall 40 cm are lost? Please clarify or remove this argument.

P4. L29-30: Were samples kept anoxic during handling and centrifugation?

P6. L12: Were multicores also collected on the same cruise from the same sites? This should be included in the section 2.1, or if the multicores came from somewhere else, tell us where.

P6. L16: How was this carbonate calculation actually done?

Results

P7. L32: Is Cu really associated with Mn? I thought it was more associated with sulfur phases and organic matter. Providing references for this and the other trace metal associations would be helpful.

Section 3.3: Mn, Co, and Cu are highly redox sensitive, so it perhaps it makes sense to combine this section with Section 3.4.

P8. L17-18. The previous sentence states that As could not be measured in the solid

phase, yet this sentence describes "considerable peaks in the solid phase and pore water concentrations of U, Mo, and As. . ."

Discussion

P9. L13: "while few are enriched" revise to "while a few are enriched"

P9. L27-35: How are authigenic and biogenic Ba distinguished? Couldn't an elevated Ba/Al ratio could be generated through either mechanism?

P9. L40: What does Ba/Al tell you anything about sedimentation rate? I am unfamiliar with this proxy, but if it is already established then perhaps an explanation in the manuscript is not needed and a good reference for the proxy would suffice.

P10. L7-14. Do you have data other than the color change to support these geochemical interpretations? For example, I would be hesitant to assume that there is no Fe(III) reduction just based on a color change. If you have solid or pore water Fe data to support this interpretation, please include it here.

P10. L21-24: "The dissolving nodules were found in the suboxic parts of the cores, as well as the brown patches inside the green sediment layers (e.g. DEA Black Patch-497 cm and DEA Trough-585 cm). The latter might be remnants of dissolving nodules. . ." The logic of this sentence is unclear. It sounds like the dissolving nodules were found in the brown patches, but I think you meant that the brown patches were found in the suboxic parts of the core. Additionally, it would be better to clarify what "the latter" are. I assumed it was the brown patches, but I'm not certain.

P10. L24: It may be more helpful for the reader if "green sediment" is referred to as "Fe(II)-rich sediment" instead.

P10. L29-31: Quotation marks are unnecessary. Much better to rephrase in your own words and just refer to source in citations.

P11. L6-21: This all seems like results; there is no interpretation of the data here, just

description. What do the upper and lower "sections" represent? Changes in diagenetic processes? Past shifts in sediment provenance? Something else? Discuss the answers to these questions here, and move the reporting of the data to the Results section.

P12. L16-17: Why is it important to understand the REY-controlling phases in the sediment? Perhaps to allow for better use of REY as indicators or proxies for certain sediment sources or diagenetic processes?

P12. L27-28: Is the ambient pore water REY are equivalent to seawater, i.e. the REY enter the sediment through diffusion?

P13. L23: "Both cores, DEA Black Patch and Reference East, are located"... → Both DEA Black Patch and Reference East are located..." In the preceding paragraph, only Reference East was discussed. The way it is written, it sounds like both cores were just discussed.

P13. L27-29. It looks to me like Reference East is almost certainly anoxic. Nitrate is consumed at a shallow depth and this site has the highest concentrations of dissolved Mn in the deep sediment. It is totally possible for sediments with a lower POC content to be anoxic. Could these trace metal content peaks in Reference East be due to a buried nodule-rich layer that is dissolving, as you have suggested elsewhere?

P13. L35: "They get preserved" revise to "They are preserved"

P13. L37: Is there a reference for the claim that turbidites are not common in the area?

P13. L27: How exactly can the influence of dissolving nodules be distinguished from hydrothermal input? Maybe with REY or trace metal ratios?

Conclusions

P14. L16: "With respect to deep-sea mining, the results show, how variable..." → "With respect to deep-sea mining, the results show how variable..." Incorrect comma

usage.

P14. L23: Again, what are the halos?

General: - I would prefer a more thorough discussion of the differences or similarities between the DISCOL and undisturbed sites in the Conclusions (if the frame of the paper is changed as I suggested above). - The discussion of the effects of the nodules on local trace metal contents should be more fleshed out here, as well. That is a particularly interesting finding of this study, in my opinion, and worth highlighting more specifically here. For example, instead of generally noting "significant small-scale differences in the mineralogical and chemical composition of sediment cores" in the final paragraph, the specific differences (enrichments in solid and pore water trace metals, difference REY signatures, etc) can be re-stated and summarized here.

Figures

Please use consistent markers for each core in all figures. For example, sometimes Reference East is represented by an empty triangle, sometimes by a filled triangle. Also, I recommend using different colors for each site, rather than shades of gray and green.

---

## Author Comment (AC1) · 13 Nov 2019

Referee comment: The authors use the variability of redox sensitive metals, organic matter, and REY between several distinct depositional environments in the Peru Basin to demonstrate that a given core may not be representative of a bigger area. While definitely true, I feel the authors over-sell a broad 'small-scale heterogeneity' emphasis while underplaying the fact that the sites were deliberately chosen in different depositional environments (including the title). Nuanced, but this data set clearly shows the importance of considering depositional environment when selecting sites and the authors don't really articulate that point, instead the message comes across as a more dismissive 'they're all different anyways.'

Authors' response: We thank the reviewer for this overall comment. We would like to emphasize that we did not chose all sites specifically due to different depositional environments. Three sites are reference sites for the DISCOL experimental area and were chosen as a comparison for the disturbance in accordance with the plowing experimental set-up. This information is added in the sampling section. The sites "black patch", "trough", and "small crater" were additionally added as potentially different sites as identified during the cruise. Overall, most sites were not chosen due to different depositional environments and especially the scale of these differences could not be expected during sampling.

Authors' changes in the manuscript: Sampling area and methods "Three cores were sampled in reference sites (South, West, East) of this experimental set-up, which are spread around the DEA within ca. 80 m difference in water depth."

Conclusion "The importance of small variations in depositional environments has been underestimated in the deep-sea and this study showed, how extensive the effects of depositional area can be on the various geochemical parameters presented here."

Referee comment: In presenting this data set, the authors also use solid and fluid phase measurements to address the sediment phases controlling the REY of the sediments. The authors argue that Fe-rich clay mineral and phosphates are the controlling phases based on Fe/Nd, Fe/Al, Fe/P, and P/Ca correlations. I believe the authors are likely correct on this assumption, but am confused as to the lack of recent citations in support of this conclusion.

Authors' response: Newer citations supporting the association of REY with clay minerals (Abbott et al., 2019 Zhang et al., 2016) and Ca phosphates (Deng et al., 2017, Kon et al., 2014, Kashiwabara et al., 2018, Liao et al., 2019) were added.
Abbott, A. N., Löhr, S. and Trethewy, M.: Are Clay Minerals the Primary Control on the Oceanic Rare Earth Element Budget?, Front. Mar. Sci., 6, doi:10.3389/fmars.2019.00504, 2019.

Deng, Y., Ren, J., Guo, Q., Cao, J., Wang, H. and Liu, C.: Rare earth element geochemistry characteristics of seawater and porewater from deep sea in western Pacific, Sci. Rep., 7(16539), 1–13, doi:10.1038/s41598-017-16379-1, 2017. Kashiwabara, T., Toda, R., Nakamura, K., Yasukawa, K., Fujinaga, K., Kubo, S., Nozaki, T., Takahashi, Y., Suzuki, K. and Kato, Y.: Synchrotron X-ray spectroscopic perspective on the formation mechanism of REY-rich muds in the Pacific Ocean, Geochim. Cosmochim. Acta, 240, 274–292, doi:10.1016/j.gca.2018.08.013, 2018.

Kon, Y., Hoshino, M., Sanematsu, K., Morita, S., Tsunematsu, M., Okamoto, N., Yano, N., Tanaka, M. and Takagi, T.: Geochemical characteristics of apatite in heavy REErich Deep-Sea Mud from Minami-Torishima Area, Southeastern Japan, Resour. Geol., 64(1), 47–57, doi:10.1111/rge.12026, 2014.

Liao, J., Sun, X., Li, D., Sa, R., Lu, Y., Lin, Z., Xu, L., Zhan, R., Pan, Y. and Xu, H.: New insights into nanostructure and geochemistry of bioapatite in REE-rich deep-sea sediments: LA-ICP-MS, TEM, and Z-contrast imaging studies, Chem. Geol., 512, 58–68, doi:10.1016/j.chemgeo.2019.02.039, 2019.

Zhang, L., Algeo, T. J., Cao, L., Zhao, L., Chen, Z. Q. and Li, Z.: Diagenetic uptake of rare earth elements by conodont apatite, Palaeogeogr. Palaeoclimatol. Palaeoecol., 458, 176–197, doi:10.1016/j.palaeo.2015.10.049, 2016.

Referee comment: The only aspect that was a bit challenging to follow was the jumps between solid phase and fluid phase, mainly as the approach to this wasn't consistent (e.g. 3.2 and 3.3 split up solid and fluid phases, then 3.4 was solid and fluid)

Authors' response: We thank the reviewer for this comment. We harmonized the structure of presenting solid phase and pore water and combined previous sections 3.2 and
3.3 to fit with the approach in previous section 3.4.

Referee comment: P1 L31 "only a small part" – has this been quantified? i.e. more impactful if you can at least assign an order of magnitude to it (0.1%? 10%?)

Authors' response: Quantification has been attempted in the publication by Ramirez-Llodra et al., 2010, where they estimated 0.01%

Authors' changes in the manuscript: Used to read: "Of this large area, only a small part has been investigated so far, resulting in a scarce dataset." Now reads: "Less than 0.01% has however been sampled and investigated in detail so far (Ramirez-Llodra et al., 2010), resulting in a scarce dataset."

Ramirez-Llodra, E., Brandt, A., Danovaro, R., De Mol, B., Escobar, E., German, C. R., Levin, L. A., Martinez Arbizu, P., Menot, L., Buhl-Mortensen, P., Narayanaswamy, B. E., Smith, C. R., Tittensor, D. P., Tyler, P. A., Vanreusel, A. and Vecchione, M.: Deep, diverse and definitely different: Unique attributes of the world's largest ecosystem, Biogeosciences, 7, 2851–2899, doi:10.5194/bg-7-2851-2010, 2010.

Referee comment: P1 L35-36 the number of pore water studies in the last decade has drastically increased but citation list heavily weighted towards earlier ones (e.g. Deng et al 2017 Scientific Reports; Kim et al 2012 Chemical Geology; Schacht et al. 2010 Journal of Geochemical Exploration; Abbott et al 2019 Frontiers in Marine Science)

Authors' response: Recent citations for relevant pore water studies added: Deng et al., 2017, Volz et al., 2018, Abbott et al., 2019, Soyol-Erdene and Huh, 2013, Kim et al., 2012, Schacht et al., 2010.

Abbott, A. N., Löhr, S. and Trethewy, M.: Are Clay Minerals the Primary Control on the Oceanic Rare Earth Element Budget?, Front. Mar. Sci., 6, doi:10.3389/fmars.2019.00504, 2019.

Deng, Y., Ren, J., Guo, Q., Cao, J., Wang, H. and Liu, C.: Rare earth element geochemistry characteristics of seawater and porewater from deep sea in western Pacific, Interactive comment

Sci. Rep., 7(16539), 1–13, doi:10.1038/s41598-017-16379-1, 2017.

Kim, J.-H., Torres, M. E., Haley, B. A., Kastner, M., Pohlman, J. W., Riedel, M. and Lee, Y.-J.: The effect of diagenesis and fluid migration on rare earth element distribution in pore fluids of the northern Cascadia accretionary margin, Chem. Geol., 291, 152–165, doi:10.1016/j.chemgeo.2011.10.010, 2012.

Schacht, U., Wallmann, K. and Kutterolf, S.: The influence of volcanic ash alteration on the REE composition of marine pore waters, J. Geochemical Explor., 106(1–3), 176–187, doi:10.1016/j.gexplo.2010.02.006, 2010.

Soyol-Erdene, T. O. and Huh, Y.: Rare earth element cycling in the pore waters of the Bering Sea Slope (IODP Exp. 323), Chem. Geol., 358, 75–89, doi:10.1016/j.chemgeo.2013.08.047, 2013.

Volz, J. B., Mogollón, J. M., Geibert, W., Martínez Arbizu, P., Koschinsky, A. and Kasten, S.: Natural spatial variability of depositional conditions, biogeochemical processes and element fluxes in sediments of the eastern Clarion-Clipperton Zone, Pacific Ocean, Deep. Res. Part I Oceanogr. Res. Pap., 140(August), 159–172, doi:10.1016/j.dsr.2018.08.006, 2018.

Referee comment: P2 L20-21 (P3) again seems like a gap in including recent literature other than the author's own 2019 paper

Authors' response: Kashiwabara et al., 2018 and Liao et al., 2019 added.

Kashiwabara, T., Toda, R., Nakamura, K., Yasukawa, K., Fujinaga, K., Kubo, S., Nozaki, T., Takahashi, Y., Suzuki, K. and Kato, Y.: Synchrotron X-ray spectroscopic perspective on the formation mechanism of REY-rich muds in the Pacific Ocean, Geochim. Cosmochim. Acta, 240, 274–292, doi:10.1016/j.gca.2018.08.013, 2018.

Liao, J., Sun, X., Li, D., Sa, R., Lu, Y., Lin, Z., Xu, L., Zhan, R., Pan, Y. and Xu, H.: New insights into nanostructure and geochemistry of bioapatite in REE-rich deep-sea sediments: LA-ICP-MS, TEM, and Z-contrast imaging studies, Chem. Geol., 512, 58–
68, doi:10.1016/j.chemgeo.2019.02.039, 2019.

Referee comment: P2 L30 (P3) ".. their detrital orgin." reference?

Authors' response: Marchig et al., 2001 (as reference for detrital origin of kaolinite and illite) and Tostevin et al., 2016, Cullers et al., 1975, Prudêncio et al., 1989 (as reference for flat REY pattern of clays) added.

Cullers, R. L., Chaudhuri, S., Arnold, B., Lee, M. and Wolf, C. W.: Rare earth distributions in clay minerals and in the clay-sized fraction of the Lower Permian Havensville and Eskridge shales of Kansas and Oklahoma, Geochim. Cosmochim. Acta, 39(12), 1691–1703, doi:10.1016/0016-7037(75)90090-3, 1975.

Marchig, V., Von Stackelberg, U., Hufnagel, H. and Durn, G.: Compositional changes of surface sediments and variability of manganese nodules in the Peru Basin, Deep. Res. Part II Top. Stud. Oceanogr., 48(17–18), 3523–3547, doi:10.1016/S0967-0645(01)00055-8, 2001.

Prudêncio, M. I., Figueiredo, M. O. and Cabral, J. M. P.: Rare earth distribution and its correlation with clay mineralogy in the clay-sized fraction of Cretaceous and Pliocene sediments (central Portugal), Clay Miner., 24(1), 67–74, doi:10.1180/claymin.1989.024.1.06, 1989.

Tostevin, R., Shields, G. A., Tarbuck, G. M., He, T., Clarkson, M. O. and Wood, R. A.: Effective use of cerium anomalies as a redox proxy in carbonate-dominated marine settings, Chem. Geol., 438, 146–162, doi:10.1016/j.chemgeo.2016.06.027, 2016.

Referee comment: In terms of REE patterns/Ce anomalies: Kang et al. 2014 Journal of Asian Earth Sciences; Kon et al. 2014, Resource Geology

Authors' response: Kon et al., 2014 and other newer references for phase association and REY patterns (Abbott et al., 2019. Zhang et al., 2016, Deng et al, 2017, Kashi-wabara et al., 2018; Liao et al., 2019) added in the discussion section
Abbott, A. N., Löhr, S. and Trethewy, M.: Are Clay Minerals the Primary Control on the Oceanic Rare Earth Element Budget?, Front. Mar. Sci., 6, doi:10.3389/fmars.2019.00504, 2019.

Deng, Y., Ren, J., Guo, Q., Cao, J., Wang, H. and Liu, C.: Rare earth element geochemistry characteristics of seawater and porewater from deep sea in western Pacific, Sci. Rep., 7(16539), 1–13, doi:10.1038/s41598-017-16379-1, 2017.

Kashiwabara, T., Toda, R., Nakamura, K., Yasukawa, K., Fujinaga, K., Kubo, S., Nozaki, T., Takahashi, Y., Suzuki, K. and Kato, Y.: Synchrotron X-ray spectroscopic perspective on the formation mechanism of REY-rich muds in the Pacific Ocean, Geochim. Cosmochim. Acta, 240, 274–292, doi:10.1016/j.gca.2018.08.013, 2018.

Kon, Y., Hoshino, M., Sanematsu, K., Morita, S., Tsunematsu, M., Okamoto, N., Yano, N., Tanaka, M. and Takagi, T.: Geochemical characteristics of apatite in heavy REE-rich Deep-Sea Mud from Minami-Torishima Area, Southeastern Japan, Resour. Geol., 64(1), 47–57, doi:10.1111/rge.12026, 2014.

Liao, J., Sun, X., Li, D., Sa, R., Lu, Y., Lin, Z., Xu, L., Zhan, R., Pan, Y. and Xu, H.: New insights into nanostructure and geochemistry of bioapatite in REE-rich deep-sea sediments: LA-ICP-MS, TEM, and Z-contrast imaging studies, Chem. Geol., 512, 58–68, doi:10.1016/j.chemgeo.2019.02.039, 2019.

Zhang, L., Algeo, T. J., Cao, L., Zhao, L., Chen, Z. Q. and Li, Z.: Diagenetic uptake of rare earth elements by conodont apatite, Palaeogeogr. Palaeoclimatol. Palaeoecol., 458, 176–197, doi:10.1016/j.palaeo.2015.10.049, 2016.

Referee comment: P4 section 2.1 unclear from the next the number of gravity cores taken and their depositional environments (with the exception of the description of the volcanic crater)

Authors' response: Number of sampled gravity cores added (7). The depositional environment for Trough and Black Patch are also explained. Since the other sites were
not chosen due to their depositional environments but due to their location with respect to the plowing experiment (within the DISCOL experimental area or in reference sites), we did not classify a depositional environment. We, however, clarified this sampling strategy, as already mentioned in a comment above. The complete overview of the cores was also given in Table 1, which we refer to in the text for further information.

Authors' changes in the manuscript: Used to read: "Samples were collected during RV SONNE cruise SO242/1 in 2015 with a gravity corer (GC) (Greinert, 2015)." Now reads: "Samples were collected from seven gravity cores (GC) during RV SONNE cruise SO242/1 in 2015 to the Peru Basin (Greinert, 2015)." Also added: "Three cores were sampled in reference sites (South, West, East) of this experimental set-up, which are spread around the DEA within ca. 80 m difference in water depth."

Referee comment: P4 L25-32 Were the cores sampled in ambient air? How long did the split cores sit before analyses? How are concerns about oxidising pore fluid before centrifugation addressed if sampled in ambient air?

Authors' response: Yes, they were sampled in ambient air but sampled immediately after splitting the core as is standard procedure. Oxygen penetration into the sediment is too slow to affect the signal significantly and our experience with sensitive variables such as Fe2+ and H2S supports this. Additionally, centrifuge vials were fully filled to minimize the oxygen content during centrifugation. Also, GC data compares well with MUC data (Paul et al., 2018) for which sampling was done in a glove bag.

Authors' changes in the manuscript: Used to read: "Once on deck, GCs were cut into 1-m sections and then divided into a working and an archive half. Working halves were immediately transported to the cold room (approx. 4°C), while the counterparts were stored as archive halves. After visual inspection, samples were collected in layers of different color, roughly one to two per meter, and transferred with plastic spoons into 50 mL acid pre-cleaned centrifuge tubes."

Now reads: "Once on deck, GCs were cut into 1 m sections and then divided into a

**BGD**
working and an archive half. Working halves were instantly transported to the cold room (approx.  $4^{\circ}$ C), while the counterparts were stored as archive halves. Samples were immediately collected to minimize contact with ambient air and thereby oxidation of reduced species in suboxic sections of the cores. After visual inspection, sediment was sampled in layers of different color, roughly one to two per meter, and transferred with plastic spoons into 50 mL acid pre-cleaned centrifuge tubes. Gravity core subsampling in ambient air is standard procedure and has been carried out regularly in previous studies (see e.g., Haeckel et al., 2001; Volz et al., 2018). Einstein-Smoluchowski informs us that diffusion will carry solutes, such as O2, only over a distance of 3 mm in 2 hours. Hence, our sampling after splitting of the core is guick enough to ensure an almost pristine signal. Our experience with more sensitive variables, such as H2S and Fe2+, supports this. The significant loss of dissolved constituents by oxidation is therefore not expected in the few hours of sampling, especially when sampling in low temperature conditions (for Mn(II) see e.g., Schnetger and Dellwig, 2012). Data for other redox-sensitive elements, e.g. U, Mo, V. As, compare well with pore water data from multicores from these sites, which were sampled in glove bags (Paul et al., 2018). Centrifuge tubes were completely filled to minimize the oxygen content during centrifugation."

Haeckel, M., König, I., Riech, V., Weber, M. E. and Suess, E.: Pore water profiles and numerical modelling of biogeochemical processes in Peru Basin deep-sea sediments ., Deep. Res. Part II Top. Stud. Oceanogr., 48, 3713–3736, doi:10.1016/S0967-0645(01)00064-9, 2001.

Paul, S. A. L., Gaye, B., Haeckel, M., Kasten, S. and Koschinsky, A.: Biogeochemical Regeneration of a Nodule Mining Disturbance Site: Trace Metals, DOC and Amino Acids in Deep-Sea Sediments and Pore Waters, Front. Mar. Sci., 5(April), 1–17, doi:10.3389/fmars.2018.00117, 2018.

Schnetger, B. and Dellwig, O.: Dissolved reactive manganese at pelagic redoxclines (part I): A method for determination based on field experiments, J. Mar. Syst., 90,

**BGD**
23-30, doi:10.1016/j.jmarsys.2011.08.006, 2012.

Volz, J. B., Mogollón, J. M., Geibert, W., Martínez Arbizu, P., Koschinsky, A. and Kasten, S.: Natural spatial variability of depositional conditions, biogeochemical processes and element fluxes in sediments of the eastern Clarion-Clipperton Zone, Pacific Ocean, Deep. Res. Part I Oceanogr. Res. Pap., 140(August), 159–172, doi:10.1016/j.dsr.2018.08.006, 2018.

Referee comment: P4 L30 Filter size is important – many of the other work compared to is 0.45 micron – not necessarily a problem, just needs to be explicitly recognized

Authors' response: We added that in the past  $0.45 \mu m$  was also often used

Authors' changes in the manuscript: We added: "In the past, sometimes 0.45  $\mu$ m filters were used, e.g., in studies by Beck et al. (2008) and Shaw et al. (1990) to which we also compare the dissolved concentrations but this intercomparison is frequently done and no significant differences have been noticed so far."

Beck, M., Dellwig, O., Schnetger, B. and Brumsack, H. J.: Cycling of trace metals (Mn, Fe, Mo, U, V, Cr) in deep pore waters of intertidal flat sediments, Geochim. Cosmochim. Acta, 72(12), 2822–2840, doi:10.1016/j.gca.2008.04.013, 2008.

Shaw, T. J., Gieskes, J. M. and Jahnke, R. A.: Early diagenesis in differing depositional environments: The response of transition metals in pore water, Geochim. Cosmochim. Acta, 54(5), 1233–1246, doi:10.1016/0016-7037(90)90149-F, 1990.

Referee comment: P5 L1 Incomplete digestions not mentioned again- need to be discussed in terms of implications

Authors' response: Sample comparison with other labs yielded comparable results, therefore the black particles should not significantly affect the results.

Authors' changes in the manuscript: We added: "Method comparison with other geochemistry labs at the beginning of this project showed that the black particles do not

**BGD**
affect the final results. Samples digested using the method above compared well with samples using HCl, HNO3, and HF and/or a microwave digestion system and those samples did not show black particles."

Referee comment: P5 L4 1-2 sentences explaining what the apex Q does would help your reader rather than having to go to the ESI website

Authors' response: We added more information on the apex Q.

Authors' changes in the manuscript: Now reads: "The desolvation nebulizer introduces the sample solution into a heated spray chamber and subsequently into a cooled condenser The apex Q thereby enhances sample introduction efficiency to decrease background noise and to increase sensitivity."

Referee comment: P8 section 3.5 this section is far less detailed than the precedent set in the earlier sections, stay consistent. Variability? Trends?

Authors' response: Now section 3.4 (used to be 3.5) REY information consolidated in this section, taken from 3.2 and 4.4 as suggested by Reviewer 2.

Authors' changes in the manuscript: Now reads: "3.4 REY profiles and patterns: solid phase and pore water Like Fe and P, REY concentrations increase with depth, especially at Reference West and DEA West (Fig. 6), and except for Small Crater. The sum of REY concentrations varies between approx. 180 ppm and 550 ppm (not shown). The buried nodules at Reference West, DEA Trough, and Reference East show similar to slightly lower REY concentrations than the sedimentary REY (see Nd in Fig. 6). Too little pore water data is available to make statements about the concentration trend with depth. [...] All cores, except Small Crater, can be divided into an upper and a lower section based on the REY concentration increase, increase in Fe/Al ratios, and a decrease of CeSN/CeSN\* ratios: Reference West and DEA West at 4.5 m, Reference South, DEA Black Patch and DEA Trough at 6 m, and Reference East at 8 m (Fig. 9). The Fe/Al ratios remain steady in the Small Crater core, as well as the negative CeSN

BGD
anomaly. The first three above mentioned cores (Reference West, DEA West, Reference South) also have higher Y/Ho and LaSN/PrSN ratios in their lower parts. The concentration increase is associated with the bottom of the green layer in cores Reference South, DEA Black Patch, DEA Trough, and Reference East. In Reference West and DEA West, where no green layer exists, the concentration increase correlates with the color change from tan to dark brown at approx. 4.5 m and the increasing Fe and P concentrations at the corresponding depth. REY are most abundant, where a higher percentage of Fe(II) in the clay minerals prevails (Reference West and DEA West)."

Referee comment: P9 section 4.1 one more sentence here reminding readers of the depth of the sites 4125 and 4216 when discussing modern CCD would be helpful

Authors' response: We added this information as requested.

Authors' changes in the manuscript: Used to read: "The present CCD is located approximately between 4200 and 4250 m water depth (Weber et al., 2000)". Now reads: "The present CCD is located approximately between 4200 and 4250 m water depth (Weber et al., 2000), slightly deeper than the water depths of the GCs presented here (4125-4208 m)."

Referee comment: P9 section 4.2 again seems lack recent references (e.g. Baldermann et al 2015 Nature Geoscience; Dong et al 2009 American Mineralogist; Huggett et al 2017 Clay Minerals)

Authors' response: Dong et al., 2009 added in Introduction 1.3 Fe-rich clay minerals Dong et al., 2009 and Baldermann et al., 2015 added in section 4.2

ć Baldermann, A., Warr, L. N., Letofsky-Papst, I. and Mavromatis, V.: Substantial iron sequestration during green-clay authigenesis in modern deep-sea sediments, Nat. Geosci., 8(11), 885–889, doi:10.1038/ngeo2542, 2015. ć Dong, H., Jaisi, D. P., Kim, J. and Zhang, G.: Microbe-clay mineral interactions, Am. Mineral., 94, 1505–1519, doi:10.2138/am.2009.3246, 2009.

BGD
Referee comment: P11 section 4.4 discussion should frame results in context of similar observations in the literature e.g. lots of work showing Nd and Fe relationship

Authors' response: We thank the reviewer for this comment. A lot of references were added throughout section 4.4, for information of clay-REY correlations, phosphate-REY correlations, uptake of REY from pore water by phosphates through coupled substitution.

Abbott, A. N., Löhr, S. and Trethewy, M.: Are Clay Minerals the Primary Control on the Oceanic Rare Earth Element Budget?, Front. Mar. Sci., 6, doi:10.3389/fmars.2019.00504, 2019.

Cullers, R. L., Chaudhuri, S., Arnold, B., Lee, M. and Wolf, C. W.: Rare earth distributions in clay minerals and in the clay-sized fraction of the Lower Permian Havensville and Eskridge shales of Kansas and Oklahoma, Geochim. Cosmochim. Acta, 39(12), 1691–1703, doi:10.1016/0016-7037(75)90090-3, 1975.

Deng, Y., Ren, J., Guo, Q., Cao, J., Wang, H. and Liu, C.: Rare earth element geochemistry characteristics of seawater and porewater from deep sea in western Pacific, Sci. Rep., 7(16539), 1–13, doi:10.1038/s41598-017-16379-1, 2017.

Elderfield, H., Hawkesworth, C. J., Greaves, M. J. and Calvert, S. E.: Rare earth element geochemistry of oceanic ferromanganese nodules and associated sediments, Geochim. Cosmochim. Acta, 45(4), 513–528, doi:10.1016/0016-7037(81)90184-8, 1981.

Jarvis, I., Burnett, W. C., Nathan, Y., Almbaydin, F. S. M., Attia, A. K. M., Castro, L. N., Flicoteaux, R., Hilmy, M. E., Husain, V., Qutawnah, A. A., Serjani, A. and Zanin, Y. N.: Phosphorite geochemistry: State-of-the-art and environmental concerns, Eclogae Geol. Helv., 87(3), 643–700, 1994.

Kashiwabara, T., Toda, R., Nakamura, K., Yasukawa, K., Fujinaga, K., Kubo, S., Nozaki, T., Takahashi, Y., Suzuki, K. and Kato, Y.: Synchrotron X-ray spectroscopic per-
spective on the formation mechanism of REY-rich muds in the Pacific Ocean, Geochim. Cosmochim. Acta, 240, 274–292, doi:10.1016/j.gca.2018.08.013, 2018.

Kon, Y., Hoshino, M., Sanematsu, K., Morita, S., Tsunematsu, M., Okamoto, N., Yano, N., Tanaka, M. and Takagi, T.: Geochemical characteristics of apatite in heavy REErich Deep-Sea Mud from Minami-Torishima Area, Southeastern Japan, Resour. Geol., 64(1), 47–57, doi:10.1111/rge.12026, 2014.

Liao, J., Sun, X., Li, D., Sa, R., Lu, Y., Lin, Z., Xu, L., Zhan, R., Pan, Y. and Xu, H.: New insights into nanostructure and geochemistry of bioapatite in REE-rich deep-sea sediments: LA-ICP-MS, TEM, and Z-contrast imaging studies, Chem. Geol., 512, 58–68, doi:10.1016/j.chemgeo.2019.02.039, 2019.

Prudêncio, M. I., Figueiredo, M. O. and Cabral, J. M. P.: Rare earth distribution and its correlation with clay mineralogy in the clay-sized fraction of Cretaceous and Pliocene sediments (central Portugal), Clay Miner., 24(1), 67–74, doi:10.1180/claymin.1989.024.1.06, 1989.

Rønsbo, J. G.: Coupled substitutions involving REEs and Na and Si in apatites in alkaline rocks from Ilímaussaq intrusion, South Greenland, and the petrological implications, Am. Mineral., 74, 896–901, doi:10.4103/0019-5545.140618, 1989.

Tostevin, R., Shields, G. A., Tarbuck, G. M., He, T., Clarkson, M. O. and Wood, R. A.: Effective use of cerium anomalies as a redox proxy in carbonate-dominated marine settings, Chem. Geol., 438, 146–162, doi:10.1016/j.chemgeo.2016.06.027, 2016.

Zhang, L., Algeo, T. J., Cao, L., Zhao, L., Chen, Z. Q. and Li, Z.: Diagenetic uptake of rare earth elements by conodont apatite, Palaeogeogr. Palaeoclimatol. Palaeoecol., 458, 176–197, doi:10.1016/j.palaeo.2015.10.049, 2016.

Referee comment: P11 L28-30 "Even though it is... Fe-rich clay minerals." this sentence is a bit awkward to read but makes one of the main points, worth reworking into 1-3 clear sentences instead of a complicated one.

BGD
Authors' response: Rewritten, see below.

Authors' changes in the manuscript: Used to read: "Even though it is unclear why only part of the core shows a correlation of AI with Nd and Fe and especially why this is once the upper and once the lower section, it corroborates the association of REY with Fe-rich clay minerals." Now reads: "It is unclear why only part of each core shows a correlation of AI with Nd and Fe and it is especially unclear why this is once the upper and once the lower core section. Nevertheless, this finding corroborates the association of REY with Fe-rich clay minerals."

Referee comment: P11 L34 'reported for nontronites." By who? Reference

Authors' response: Reported by Alt, 1988; Mascarenhas-Pereira and Nath, 2010; Murnane and Clague, 1983 as shown and referenced in Fig. 9., References added here in text as well.

Referee comment: P12 L11, 27-28 This jumps from the last thought- where is the pore water getting the REY and how do they compare to the sediments? Needs a few more sentences walking the reader through the logic.

Authors' response: We thank the reviewer for pointing this out. We added some more information on potential sources of REY to the pore water and that the pattern is taken up by Ca phosphates through coupled substitution of REE3+ and e.g., Na+ for Ca2+. The paragraphs were also rearranged so that the logic should make more sense now.

Referee comment: P13 sections 4.5-4.6 hard to track dissolved versus solid- separate clearly or consistently label

Authors' response: Labels adjusted.

Authors' changes in the manuscript: Now reads: 4.5 Dissolved Mn, Co, and Cu 4.6 Redox-sensitive metals Mo, U, As, and V: solid phase and pore water

Referee comment: P13 L11 "They display: : : in oxic waters" is this a general statement
or also observed at these Stations

Authors' response: This is a general statement. No oxic data presented here. But reference to the oxic MUCs added and some more interpretation of the suboxic conservative profiles.

Authors' changes in the manuscript: Added: "In the suboxic sediments presented here, profiles are largely conservative (Fig. 8) except few peaks and in the same range as concentrations in oxic pore waters in the Peru Basin (Paul et al., 2018). Therefore, conditions in the Peru Basin sediments are likely insufficiently reducing to lead to a redox change for these elements with depth."

Referee comment: P13 L28 is the mineralogy of the grey bands known?

Authors' response: Unfortunately, the mineralogy of the gray bands is not known.

Referee comment: P14 L7-10 This is a very fair description of the spatial heterogeneity in terms of the different depositional environments

Authors' response: We thank the reviewer for this comment.

Referee comment: Fig 1) the red text is hard to read

Authors' response: Red text changed to black.

Figure 1: The Peru Basin with location of the DISCOL area. The map was created using GeoMapApp (www.geomapapp.org), CC BY, and its integrated default basemap Global Multi-Resolutional Topography (GMRT), CC BY (Ryan et al., 2009).

Referee comment: Table 1) why is 84GC3 different significant figures for water depth then the rest of the sites?

Authors' response: Harmonized, all without decimal places.

Referee comment: Figure 2) Can the color bar be larger- very hard to read the text in the key
Authors' response: Figure was remade. We acknowledge QGIS and thank Anne Hennke and Jens Greinert for the bathymetry data in the figure caption.

Figure 2: GC sampling locations in the Peru Basin. The circle indicates the DISCOL experimental area (DEA) that was traversed with a plow harrow. Created with QGIS with bathymetry data provided by Anne Hennke and Jens Greinert, DSM group, GEOMAR.

Referee comment: Figure 9) color coding on this figure is great- makes it really easy to follow what is going on!

Authors' response: We thank the reviewer for this comment.

Referee comment: Figure 11) is this the same key as figure 12?

Authors' response: We thank the reviewer for spotting the missing legend. It generally is the same as figure 12, but the legend was now added to figure 11 as well.

---

## Author Comment (AC2) · 13 Nov 2019

Referee comment: However, I do have some broad concerns about the use and presentation of the data, consideration of confounding variables, and the general frame of the paper. My largest concern is the fact that half the sites investigated in this study were subject to a disturbance and recolonization experiment thirty years ago, while the other half are pristine. In my opinion, there is not enough consideration of this potentially confounding variable, and how the impacts of ploughing could have caused some of the observed heterogeneity in the sediment. This is not to say the DISCOL

experiment invalidates the results of this study; in fact, I think that a greater focus on the differences (or lack thereof) between the DEA and undisturbed sites would make a much more compelling frame for the paper. Additionally, the paper would be greatly strengthened by a more thorough discussion of how the results of this study, as a long term follow-up to the DISCOL experiment, relates to deep sea nodule mining and could inform future mining decisions. On the other hand, if the authors feel this study does not have a strong connection to current mining activities and decisions, then this should not be mentioned (e.g. Page 2, Line 12 and Page 14, Line 16), as the connection may mislead readers.

Authors' response: We thank the reviewer for this comment. We would like to point out, however, that the focus of this manuscript was the natural variability and related to that implications that could help to inform deep-sea mining decisions. A direct impact from mining is only expected in the upper ca. 20 cm of sediment and these impacts have been addressed in Paul et al., 2018 for biogeochemistry with a focus on metal cycling. The data presented here focuses on a more basic geochemical description of this site including depositional variations over time (on geological time scales) and space (e.g. the variability in redox-processes on small spatial scales). The 10 m long gravity cores (GCs) presented here are not suitable for a disturbance comparison, as it is not clear if the GCs were taken in tracks in the experimental site or next-to tracks. The experimental area is not equally disturbed. From surface sediment studies, we know that the sediment is only impacted in the surface ca. 20 cm and that the pore water metal concentrations have re-established an equilibrium after 26 years (Paul et al., 2018). We therefore do not expect disturbance related signals in the GC pore water data and that the impacted surface sediment is lost during sampling, as described in the methods section of this manuscript. Nevertheless, we think that it would not be correct to completely leave out brief background information about the experiment, because this is still the basis for sample distribution and why this site was chosen.

Referee comment: Finally (in full acknowledgement that I am not an expert in rare

earth elements), after reading the paper I was left uncertain about the usefulness and relevance of the REY data set in the frame of the study. It was unclear to me what further information the REY data imparted regarding biogeochemical processes and variation between the sites that was not apparent in the other (trace metal, carbon, etc.) data sets. This aspect of the paper could be improved by more background on REY in the introduction and a more detailed discussion of interpretation of the REY results in the frame of biogeochemical differences between the sites and/or the impact of polymetallic nodules at the surface or buried in the sediment.

Authors' response: We thank the reviewer for this comment and added more background information and discussion on REY throughout the manuscript. These changes are in line with other requests for further information from reviewer #2. The observed changes in REY depth distribution or pattern support changes in the other parameters (color, major element trends) that are sometimes subtle. In general, however, the REY are a good parameter for the interregional comparison to other deep-sea sites, e.g. the CCZ, where pore-water and solid-phase REYSN patterns are completely different, which also tells us something about the interregional variability of nodule areas with respect to deposited material, sediment composition and early diagenetic processes.

Authors' changes in manuscript: Introduction: "Fractionation can indicate particle-solution interactions in the marine environment, when for example Ce or Y are decoupled from their REY neighbors during redox cycling or hydrogenetic Mn- and/or Fe-(oxyhydr)oxide formation, respectively (Bau, 1999; Bau et al., 1997, 1998). This is because of different surface complex stabilities between the individual REY (Bau et al., 1997). The subtle differences between complex stability constants are sufficient to lead to fractionation because of preferential scavenging or mobilization of the light REY (LREY; La-Nd), middle REY (MREY; Sm-Dy), or heavy REY (HREY; Y-Lu) (Cantrell and Byrne, 1987; Elderfield, 1988)." Discussion: Subheading section 4.4 changed to: REY as indicators for variability of deep-sea sediments Information added in the discussion: "The change in REY concentration with depth could be associated with past changes

in sediment deposition – especially in cores Reference West and DEA West, where a color change from tan to dark brown is visible but no green layers. A second impact of REY concentration change might be related to a change in redox-zonation in cores Reference South, DEA Black Patch, DEA Trough, and Reference East, where the lower end of the green layers coincides with the REY concentration increase. Small changes in the REY concentrations and SN patterns can be observed that correlate with other changes, e.g. changes in major element concentration (Fe, Al, P), or color (tan, dark brown, green). Small-scale variability is therefore also visible in the REY concentrations and SN patterns within the Peru Basin. Correlations of REY and major elements help to elucidate phase associations of REY, which are important to understand before interpreting REY cycling."

Abstract Referee comment P1 L15 and L 23 Be careful to clarify whether the heterogeneity referred to is between sites or between depths at a single site. The "variability" in line 23 seems as though it is referring to Mn and Co concentration peaks with depth, rather than differences between the sites.

Authors' response: Clarified in text: L15 spatial heterogeneity, L23 both, between sites and with depth in cores.

Introduction Referee comment P1 L34 What is meant by biogeochemical heterogeneity, exactly? Different processes? Different carbon contents? Simply giving a few examples of relevant biogeochemical parameters that vary between sites would be helpful

Authors' response: Added in text, see below.

Authors' changes in manuscript: Reads now: "...biogeochemical heterogeneity with respect to e.g., sedimentation rate, POC flux, TOC contents, oxygen penetration depth, and thereby extension of the oxic and suboxic zone (Volz et al., 2018)."

Referee comment P1 L34-35 "In the past, few spread-out samples were collected for

pore-water and solid-phase geochemical analyses" As written, the sentence does not emphasize the sparse nature of past sampling. Rephrase to something like: "In the past, cores collected for pore water and solid phase geochemical analyses have been sparse and separated by large distances." I'm sure there's a better way to word that, but hopefully you understand what I mean.

Authors' response: Rephrased in text, see below.

Authors' changes in manuscript: Reads now: "Similarly, many studies in the past collected cores for pore-water and solid-phase geochemical analyses based on sparse sampling distribution and spread over large areas"

Referee comment P1 L36 "on small spatial scale" revise to "on small spatial scales."

Authors' response: Changed accordingly.

Referee comment P2 L1 "could show" revise to "showed"

Authors' response: Changed accordingly.

Referee comment P2 L2 "studies of few samples" revise to "studies of a few isolated samples" or something similar.

Authors' response: Changed accordingly.

Referee comment P2 L12 How does the heterogeneity discussed in the paper relate to deep-sea mining? Will the results help inform mining decisions? Do they imply that mining does not have a significant impact on sediment biogeochemistry? If there is not a strong connection between the results and mining, I would minimize discussion of mining except to explain the reason for the DISCOL experiment.

Authors' response: We thank the reviewer for this comment. As already mentioned in the first response, we think the introduction to the DISCOL experiment is relevant and even though we are not assessing disturbance impacts here, this baseline data is valuable information that needs to be kept in mind when planning the set-up of environ-

mental impact assessments with respect to deep-sea mining. Therefore, we would like to keep this connection in the manuscript. This is already stated in the conclusions: "With respect to deep-sea mining, the results show how variable the deep-sea floor can be and that extensive baseline studies are necessary before the onset of mining and impact analyses." We added some more aspects in the conclusions, see changes below.

Authors' changes in manuscript: Used to read: "Since the geochemical composition of the sediment, including POC content and redox conditions, has a major impact on microbial processes in the sediment and associated biological life, this small-scale heterogeneity may also be relevant for biological productivity and diversity in the deep-sea" We added: "...deep-sea, as well as biological recovery after deep-sea mining disturbances,"

Referee comment P2 L23 "Mineralogical investigations of long cores were conducted extensively" This seems to contradict the previous sentence.

Authors' response: Mineralogical investigations of the upper 10 m of sediment were conducted extensively but not geochemical analyses. We added "however" to this sentence to make the difference clearer.

Referee comment Section 1.2 and 1.3 Consider placing Section 1.3 before Section 1.2, so that the reader gets an idea of the study area before learning about previous work in the area. Learning about the sediment biogeochemistry and the presence of nodules will help the reader understand why the mining experiment occurred here.

Authors' response: We thank the reviewer for this comment. Section 1.2 was placed after section 1.5. We liked the idea to present the impact description after the description of the area, but we wanted to keep the area description (early diagenesis (1.2), Fe-rich minerals (1.3), REY (1.4)) together. Additionally, the previous work section now fits nicely before the research aim section.

Referee comment P2 L31-37 Throughout the paper, the authors rely on sediment color to make assumptions regarding the geochemical composition of the sediment. Color can be a useful indicator, but should be backed up by true geochemical data. If such data exists, please include it in this paragraph (and others discussed below). If it does not, make this clear to the reader and be transparent that some of your mineralogical assumptions are based solely on color and may not be entirely reliable. For example, in line 35: "color change typically indicates re-oxidation" or in line 34 "The Fe(III) to Fe(II) redox boundary is assumed to occur where the sediment color changes from tan to green."

Authors' response: We thank the reviewer for this comment. We partly agree, that we should transparently explain where our assumptions about color are backed up by geochemical data and where we use color as an additional indication that changes we observe in the geochemistry are also visible in color. It has been demonstrated well for the Peru Basin that the tan-green color change fits to the Fe(II)-Fe(III) redox boundary, where NO3- is completely consumed, see Lyle, 1983, König et al., 1997, 1999, where this was specifically shown for the Peru Basin and we cite these papers throughout our manuscript when we discuss the tan-green color change. Therefore, we think that this is an assumption that is quite valid. To our knowledge, other color changes, e.g. from tan to dark brown, have not been systematically geochemically analyzed so far, but we use color only to show that there is a visible change in the cores and we can see changes in the geochemical composition at the same depths.

We changed the two suggestions for P2 L34 and L35 accordingly.

Referee comment P3 L10-12 Here, I'm not certain that the sediment colors provide any useful information, since they should not be solely relied upon to determine geochemical composition later in the paper.

Authors' response: As mentioned in our response above, we think color is a valid indicator for the Fe(II)-Fe(III) redox-change. The papers we cite on P3 L10-12 all specifically studied color change in relation to mineralogy, wherefore this is a basis we can build upon and that is justified.

Referee comment P3 L18 Fractionation associated with which processes? Again, I am not a rare earths expert, so it would help me to understand what processes REY fractionation can indicate.

Authors' response: We added this information in the introduction section about REY, see below.

Authors' changes in manuscript: "Fractionation can indicate particle-solution interactions in the marine environment, when for example Ce or Y are decoupled from their REY neighbors during redox cycling or hydrogenetic Mn- and/or Fe-(oxyhydr)oxide formation, respectively (Bau, 1999; Bau et al., 1997, 1998). This is because of different surface complex stabilities between the individual REY (Bau et al., 1997). The subtle differences between complex stability constants are sufficient to lead to fractionation because of preferential scavenging or mobilization of the light REY (LREY; La-Nd), middle REY (MREY; Sm-Dy), or heavy REY (HREY; Y-Lu) (Cantrell and Byrne, 1987; Elderfield, 1988)."

Bau, M.: Scavenging of dissolved yttrium and rare earths by precipitating iron oxyhydroxide: Experimental evidence for Ce oxidation, Y-Ho fractionation, and lanthanide tetrad effect, Geochim. Cosmochim. Acta, 63(1), 67–77, 1999.

Bau, M., Möller, P. and Dulski, P.: Yttrium and lanthanides in eastern Mediterranean seawater and their fractionation during redox-cycling, Mar. Chem., 56, 123–131, doi:10.1016/S0304-4203(96)00091-6, 1997.

Bau, M., Usui, A., Pracejus, B., Mita, N., Kanai, Y., Irber, W. and Dulski, P.: Geochemistry of low-temperature water-rock interaction: Evidence from natural waters, andesite, and iron-oxyhydroxide precipitates at Nishiki-numa iron-spring, Hokkaido, Japan, Chem. Geol., 151(1–4), 293–307, doi:10.1016/S0009-2541(98)00086-2, 1998.

Cantrell, K. J. and Byrne, R. H.: Rare earth element complexation by carbonate and oxalate ions, Geochim. Cosmochim. Acta, 51(3), 597–605, doi:10.1016/0016-7037(87)90072-X, 1987.

Elderfield, H.: The Oceanic Chemistry of the Rare-Earth Elements, Philos. Trans. R. Soc. A Math. Phys. Eng. Sci., 325, 105–126, doi:10.1098/rsta.1988.0046, 1988.

Referee comment General Introduction I would like to have more background on nodules what are they composed of, how are they formed and how do they relate to the biogeochemistry of the sediment? How do the nodules "dissolve" and form the observed haloes in the sediment?

Authors' response: We added one sentence in the introduction about nodules. The dissolution of nodules is dependent on the environment where they are buried and cannot be easily summarized or generalized. The specifics of the "halos" we found in the Peru Basin sediments where the nodules dissolve are presented in the results (3.1) and discussion (4.2) and we do not have more information than is already presented.

Authors' changes to the manuscript: "Polymetallic nodules are mineral precipitates of Mn oxides and Fe (oxyhydr)oxides that form around a nucleus, e.g. bone, rock or nodule fragments, from accretion of Mn oxides and Fe (oxyhydr)oxides from seawater and pore water (Hein and Koschinsky, 2014)."

Hein, J. R. and Koschinsky, A.: Deep-Ocean Ferromanganese Crusts and Nodules, in Treatise on Geochemistry, vol. 13, pp. 273–291, Elsevier., 2014.

Referee comment General Introduction There should be consideration of the relationship of topography to sediment heterogeneity. It seems intuitive that the sediments will be heterogeneous if the topography is as varied as it is, and this is mentioned in the Conclusions but should be included in the Introduction and Discussion as well.

Authors' response: We thank the reviewer for this comment but would like to stress, that heterogeneity of and the effect of topography on deep-sea sediments has often

been underestimated in the past, as also mentioned in the introduction (section 1.1) and that this is one of the main goals of this study, to show this varied topography and the related heterogeneity in biogeochemistry (section 1.6). We discuss the relation to topography especially with respect to the lower lying cores (see examples below) and the impact of location/topography on nodule distribution/burial and organic matter deposition. We therefore think that this is represented sufficiently in the discussion.

"The cores with extensive green layers were located in depressions (DEA Trough and Reference East) and had few or no nodules on the seafloor (DEA Black Patch, DEA Trough, Reference East)." "Most buried nodules, however, were found in depressions (Table 1) suggesting that their distribution and burial might be related to bathymetry-controlled sediment depocenters." ". . . . In addition, Reference East is located at greater water depth (56-91 m deeper than the other sites)."

Referee comment General Introduction I am curious whether the sediments within the DISCOL area have the same redox zonation? Parts of the zonation must have been removed, but have they re-established since 1989? Discussion of this would help the reader understand the similarities and differences between the DISCOL sites and the undisturbed sites.

Authors' response: As there is no geochemical data available from before the 1989 disturbance experiment as mentioned in the introduction, it is difficult to say how variable the natural variability with respect to redox-zonation was within the DEA and how much variability now is based on the disturbance impact. Regeneration of the redox-zonation is beyond the scope of this manuscript and is addressed elsewhere (Paul et al., 2018; Haffert et al., in review).

Referee comment General Introduction If the authors decide to maintain deep sea mining as a part of the "implications" of this study, there should be more background in the Introduction on mining in the area, what is mined, and how?

Authors' response: We would like to keep the implications for deep-sea mining we

derive from the spatial heterogeneity in this manuscript, but as mining is not the main focus of this paper, we would not like to increase the background information about mining in the introduction. We already state that polymetallic nodules will be mined and wherever necessary in the introduction, methods, and discussion, we state how much of the sediment is expected to be impacted. As the exact mining technology is not yet present, a discussion on how nodules will be mined would be beyond the scope of this manuscript.

Methods Referee comment P4 L12 The "Therefore" is unnecessary. In fact, this sentence should go after the description of the disturbance experiment, maybe at the end of the paragraph.

Authors' response: Changed accordingly.

Referee comment P4 L20-22 I am not convinced that the ploughing had no effect on the sediment, or that the loss of sediment during coring removes that effect. The 20 cm lost from the ploughing was removed 25 years ago; the 20 cm lost in GC sampling was lost the instant the core was taken. Also, shouldn't the GC cores in the disturbed sites also lose 20 cm, so overall 40 cm are lost? Please clarify or remove this argument.

Authors' response: The surface sediment was not necessarily removed/disturbed in the DEA by plowing. Only plow tracks in the DEA are disturbed, not the entire surface in the DEA circle. Therefore, the degree of disturbance to the GCs is unclear anyways, as GCs are not sampled with video guidance and we do not know if the GC was sampled in a track or not. We added this information.

Authors' changes in manuscript: Used to read: "The plowing affected approximately the upper 20 cm of the sediment (Paul et al., 2018), which are often lost or disturbed during GC sampling so that the disturbance experiment should not affect the comparison of the GCs, regardless whether they were sampled in disturbed or undisturbed sites." Now reads: "The plowing affected approximately the upper 20 cm of the sediment in the tracks and less in areas of resettled sediment, which was determined based on

multicorer data from the DISCOL area, including the sites corresponding to the GCs presented here (Paul et al., 2018). This upper layer is often lost or disturbed during GC sampling so that the disturbance experiment should not affect the comparison of the GCs, regardless whether they were sampled in disturbed or undisturbed sites. As the GCs are not sampled with video guidance, it is unclear if a GC was taken exactly in a track or not; therefore, a comparison of disturbed and undisturbed sites is not possible based on GCs."

Referee comment P4 L29-30 Were samples kept anoxic during handling and centrifugation?

Authors' response: Samples were not kept anoxic during sampling but the O2 contact was minimized. More information was added, also in line with questions about sample handling from reviewer 1.

Authors' changes in manuscript: Used to read: "Once on deck, GCs were cut into 1 m sections and then divided into a working and an archive half. Working halves were immediately transported to the cold room (approx. 4°C), while the counterparts were stored as archive halves. After visual inspection, samples were collected in layers of different color, roughly one to two per meter, and transferred with plastic spoons into 50 mL acid pre-cleaned centrifuge tubes."

Now reads: "Once on deck, GCs were cut into 1 m sections and then divided into a working and an archive half. Working halves were instantly transported to the cold room (approx. 4°C), while the counterparts were stored as archive halves. Samples were immediately collected to minimize contact with ambient air and thereby oxidation of reduced species in suboxic sections of the cores. After visual inspection, sediment was sampled in layers of different color, roughly one to two per meter, and transferred with plastic spoons into 50 mL acid pre-cleaned centrifuge tubes. Gravity core subsampling in ambient air is standard procedure and has been carried out regularly in previous studies (see e.g., Haeckel et al., 2001; Volz et al., 2018). Einstein-Smoluchowski informs us that diffusion will carry solutes, such as O2, only over a distance of 3 mm in 2 hours. Hence, our sampling after splitting of the core is quick enough to ensure an almost pristine signal. Our experience with more sensitive variables, such as H2S and Fe2+, supports this. The significant loss of dissolved constituents by oxidation is therefore not expected in the few hours of sampling, especially when sampling in low temperature conditions (for Mn(II) see e.g., Schnetger and Dellwig, 2012). Data for other redox-sensitive elements, e.g. U, Mo, V, As, compare well with pore water data from multicores from these sites, which were sampled in glove bags (Paul et al., 2018). Centrifuge tubes were completely filled to minimize the oxygen content during centrifugation."

Referee comment P6 L12 Were multicores also collected on the same cruise from the same sites? This should be included in the section 2.1, or if the multicores came from somewhere else, tell us where.

Authors' response: Yes, multicorers were also collected and information about this was added in section 2.1 (sampling).

Authors' changes in manuscript: Now reads: "The plowing affected approximately the upper 20 cm of the sediment in the tracks and less in areas of resettled sediment, which was determined based on multicorer data from the DISCOL area, including the sites corresponding to the GCs presented here (Paul et al., 2018).

Referee comment P6 L16 How was this carbonate calculation actually done?

Authors' response: We provide the formula we used (Eq. 1) and added a link to the dataset on PANGAEA where the depths for CaCO3 and metal data can be compared, which also allows for the comparison of metal data with and without CaCO3 correction.

Authors' changes in manuscript: [metalcorrected]=[metal]/((100-[CaCO3 wt.%]) )*100 (1) For more details see https://doi.pangaea.de/10.1594/PANGAEA.903517.

Results Referee comment P7 L32 Is Cu really associated with Mn? I thought it was

more associated with sulfur phases and organic matter. Providing references for this and the other trace metal associations would be helpful.

Authors' response: We rephrased this sentence (see below). We have noticed the association of Cu with Mn in surface sediments in the Peru Basin (Paul et al., 2018) but did not want to go into this discussion in the results section here. The reference is provided in the discussion section, where we already stated that the Cu behavior is quite different from Mn and Co. "Copper does not display the west-to-east-trend in the pore water profiles and does also not show an increase at depths where Mn and Co are enriched in the suboxic zone. A deviation of Cu from the behavior of Mn, Co, Ni etc. has already been found in our previous study (Paul et al., 2018). While Mn, Co, and Ni are largely controlled by Mn oxides and their reduction during POC degradation (Heggie and Lewis, 1984; Klinkhammer, 1980; Shaw et al., 1990), Cu is largely controlled by the release from organic matter during early diagenesis and only partially due to association with Mn oxides (Klinkhammer, 1980; Shaw et al., 1990)."

We also provided information and references for other trace metal associations in the discussion (see below).

Authors' changes in manuscript: Used to read:" In these cores, concentrations of P, Nd, Mn, as well as metals associated with Mn, such as Cu, Ni, and Co, increase below 400 cm (Fig. 6)." Now reads: "In these cores, concentrations of P, Cu, Nd, Mn, as well as metals associated with Mn, such as Cu, Ni, and Co, increase below 400 cm (Fig. 6)."

Added for other trace metal associations: Used to read: "The redox-sensitive metals Mo, U, As, and V are soluble under oxic conditions and are bound to the solid phase 10 under anoxic conditions in the sediment (Beck et al., 2008; Elbaz-Poulichet et al., 1997; Wang, 2012). They display conservative type profiles in oxic waters (Beck et al., 2008). In the gray bands in Reference East, where U, Mo, V, and As concentrations peak in the solid phase and pore water, dissolved Co concentrations are low (even below the LOQ at 0.13 mg/kg) and dissolved Mn concentrations are slightly lower than in the surrounding sediment above and below (Fig. 7)." Now reads: "The redox-sensitive metals Mo, U, As, and V are soluble under oxic conditions and are bound to the solid phase under anoxic conditions in the sediment (Beck et al., 2008; Elbaz-Poulichet et al., 1997; Wang, 2012). They display conservative type profiles in oxic pore waters and are all associated with cycling of organic material, Mn (for Mo, As, V), and Fe (for U, As) (Beck et al., 2008; Telfeyan et al., 2017). In the suboxic sediments presented here, profiles are largely conservative (Fig. 8) except few peaks and in the same range as concentrations in oxic pore waters in the Peru Basin (Paul et al., 2018). Therefore, conditions in the Peru Basin sediments are likely insufficiently reducing to lead to a redox change for these elements with depth. An exception are the gray bands in Reference East, where U, Mo, V, and As concentrations peak in the solid phase and pore water, dissolved Co concentrations are low (even below the LOQ at an average of 0.14 $\mu$g/kg) and dissolved Mn concentrations are slightly lower than in the surrounding sediment above and below (Fig. 7)."

Referee comment Section 3.3 Mn, Co, and Cu are highly redox sensitive, so it perhaps it makes sense to combine this section with Section 3.4.

Authors' response: Combined with section 3.2 to keep solid phase and pore water together, as suggested by Referee #1 and to be consistent with sections 3.4 and 3.5.

Referee comment P8 L17-18 The previous sentence states that As could not be measured in the solid phase, yet this sentence describes "considerable peaks in the solid phase and pore water concentrations of U, Mo, and As: : :"

Authors' response: Rephrased: (only pore water) added for As.

Authors' changes in manuscript: Used to read: "Arsenic and Cd could not be determined in the solid phase due to the formation of gaseous AsF5 during HF digestion of the samples as well as unreliable Cd measurements with the ICP-MS, respectively. Considerable peaks in the solid phase and pore water concentrations of U, Mo, and

Interactive
comment

As are, however, visible for Reference East at depths 229.5 cm, 236.5 cm and 330 cm, where diffuse dark gray bands of approximately 1 cm thickness exist in the sediment (de Stigter, 2015)." Now reads: "Arsenic and Cd could not be determined in the solid phase due to the formation of gaseous AsF5 during HF digestion of the samples as well as unreliable Cd measurements with the ICP-MS, respectively. Considerable peaks in the solid phase and pore water concentrations of U, Mo, and As (only pore water) are, however, visible for Reference East at depths 229.5 cm, 236.5 cm and 330 cm, where diffuse dark gray bands of approximately 1 cm thickness exist in the sediment (de Stigter, 2015)."

Discussion Referee comment P9 L13 "while few are enriched" revise to "while a few are enriched"

Authors' response: Changed accordingly.

Referee comment P9 L27-35 How are authigenic and biogenic Ba distinguished? Couldn't an elevated Ba/Al ratio could be generated through either mechanism?

Authors' response: The main point here is that in the oxic/suboxic setting with little bioproductivity, authigenic barite production is unlikely and that we are therefore excluding it, not based in the ratios. This is already described and referenced in the manuscript.

Referee comment P9 L40 What does Ba/Al tell you anything about sedimentation rate? I am unfamiliar with this proxy, but if it is already established then perhaps an explanation in the manuscript is not needed and a good reference for the proxy would suffice.

Authors' response: Biogenic Ba is a bioproductivity indicator. We think in order to better understand our conclusions drawn from this ratio, it is important to briefly explain the ratio for the reader. Therefore, we would like to keep the brief explanation in the discussion.

Referee comment P10 L7-14 Do you have data other than the color change to support these geochemical interpretations? For example, I would be hesitant to assume that

there is no Fe(III) reduction just based on a color change. If you have solid or pore water Fe data to support this interpretation, please include it here.

Authors' response: As mentioned in some comments above, the combination of the tan-green color change and the nitrate penetration depth is a well developed concept for the behavior of Fe(II)/Fe(III) in the Peru Basin (see Lyle, 1983, Drodt et al., 1997, König et al., 1997, 1999). Where nitrate is consumed, Fe(III) in the clay minerals is reduced and the increasing Fe(II) content gives the sediment the green color. Since the Fe(III) is bound in clay minerals and not in Fe-oxyhydroxides, there is no mobilization of Fe into the pore water upon reduction. Pore water Fe was monitored and was never detected in any core, the detection limit being 0.5-1 $\mu$mol/L.

Authors' changes in manuscript: Used to read: "The color change from tan to green, visible in four cores (Fig. 3), represents the NO3- penetration depth and the green color results from increased Fe(II) content in the nontronite (Drodt et al., 1997; König et al., 1997, 1999; Lyle, 1983)." Now reads: "The color change from tan to green, visible in four cores (Fig. 3), represents the NO3- penetration depth and the green color results from increased Fe(II) content in the nontronite, a process that has been well established for sediments in the Peru Basin (Drodt et al., 1997; König et al., 1997, 1999; Lyle, 1983). No dissolved Fe was detected in the pore water (limit of detection 0.5-1 $\mu$mol/L), confirming that there is not Fe-oxyhydroxide reduction taking place."

Referee comment P10 L21-24 "The dissolving nodules were found in the suboxic parts of the cores, as well as the brown patches inside the green sediment layers (e.g. DEA Black Patch-497cm and DEA Trough-585 cm). The latter might be remnants of dissolving nodules: : :" The logic of this sentence is unclear. It sounds like the dissolving nodules were found in the brown patches, but I think you meant that the brown patches were found in the suboxic parts of the core. Additionally, it would be better to clarify what "the latter" are. I assumed it was the brown patches, but I'm not certain.

Authors' response: Rephrased.

Authors' changes in manuscript: Used to read: "The dissolving nodules were found in the suboxic parts of the cores, as well as the brown patches inside the green sediment layers (e.g. DEA Black Patch-497 cm and DEA Trough-585 cm). The latter might be remnants of dissolving nodules..." Now reads: "Dissolving nodules and brown patches inside the green sediment layers (e.g., DEA Black Patch-497 cm and DEA Trough-585 cm) were found in the suboxic parts of the cores. The brown patches might be remnants of dissolving nodules..."

Referee comment P10 L24 It may be more helpful for the reader if "green sediment" is referred to as "Fe(II)-rich sediment" instead.

Authors' response: Changed accordingly.

Referee comment P10 L29-31 Quotation marks are unnecessary. Much better to rephrase in your own words and just refer to source in citations.

Authors' response: Rephrased.

Authors' changes in manuscript: Used to read: "(1) "precipitation from hydrothermal fluids", (2) "alteration of volcanic rocks", and (3) "low-temperature combination of biogenic silica and" Fe (oxyhydr)oxides (Cole and Shaw, 1983, p.239)." Now reads: "(1) precipitation from hydrothermal fluids, (2) alteration of volcanic rocks, and (3) interaction of Fe (oxyhydr)oxides and biogenic silica at low temperature (Cole and Shaw, 1983)."

Referee comment P11 L6-21 This all seems like results; there is no interpretation of the data here, just description. What do the upper and lower "sections" represent? Changes in diagenetic processes? Past shifts in sediment provenance? Something else? Discuss the answers to these questions here, and move the reporting of the data to the Results section.

Authors' response: Descriptive information moved to the results, section 3.4.

Authors' changes in manuscript: Now moved to results section 3.4, consolidated and

expanded in accordance with comments from referee #1: "Like Fe and P, REY concentrations increase with depth, especially at Reference West and DEA West (Fig. 6), and except for Small Crater. The sum of REY concentrations varies between approx. 180 ppm and 550 ppm (not shown). The buried nodules at Reference West, DEA Trough, and Reference East show similar to slightly lower REY concentrations than the sedimentary REY (see Nd in Fig. 6). Too little pore water data is available to make statements about the concentration trend with depth. [. . .] All cores, except Small Crater, can be divided into an upper and a lower section based on the REY concentration increase, increase in Fe/Al ratios, and a decrease of CeSN/CeSN* ratios: Reference West and DEA West at 4.5 m, Reference South, DEA Black Patch and DEA Trough at 6 m, and Reference East at 8 m (Fig. 9). The Fe/Al ratios remain steady in the Small Crater core, as well as the negative CeSN anomaly. The first three above mentioned cores (Reference West, DEA West, Reference South) also have higher Y/Ho and LaSN/PrSN ratios in their lower parts. The concentration increase is associated with the bottom of the green layer in cores Reference South, DEA Black Patch, DEA Trough, and Reference East. In Reference West and DEA West, where no green layer exists, the concentration increase correlates with the color change from tan to dark brown at approx. 4.5 m and the increasing Fe and P concentrations at the corresponding depth. REY are most abundant, where a higher percentage of Fe(II) in the clay minerals prevails (Reference West and DEA West)."

Added in the discussion for REY: "The change in REY concentration with depth could be associated with past changes in sediment deposition – especially in cores Reference West and DEA West, where a color change from tan to dark brown is visible but no green layers. A second impact of REY concentration change might be related to a change in redox-zonation in cores Reference South, DEA Black Patch, DEA Trough, and Reference East, where the lower end of the green layers coincides with the REY concentration increase. Small changes in the REY concentrations and SN patterns can be observed that correlate with other changes, e.g. changes in major element concentration (Fe, Al, P), or color (tan, dark brown, green). Small-scale variability is therefore

also visible in the REY concentrations and SN patterns within the Peru Basin."

Referee comment P12 L16-17 Why is it important to understand the REY-controlling phases in the sediment? Perhaps to allow for better use of REY as indicators or proxies for certain sediment sources or diagenetic processes?

Authors' response: REY can be good indicators for sediment provenance or diagenetic processes in certain settings. For that, their cycling between the pore water and solid phase needs to be well understood. In the Peru Basin, the REY behave relatively coherently, small changes can be observed that correlate with other changes, e.g. changes in major element concentration (Fe, Al, P) or color (tan, dark brown, green). Therefore, variability can also be seen in the REY but they are also a good indicator for interregional comparison of sediments, e.g. comparing the Peru Basin and the CCZ, a second nodule area in the Pacific. This has largely been provided in response to the comment above.

Authors' changes in manuscript: Additionally added in the discussion: "Correlations of REY and major elements help to elucidate phase associations of REY, which are important to understand before interpreting REY cycling." "This is the same process as in the central equatorial Pacific (see e.g., Paul et al., 2019), but the pore water REYSN pattern is different in the Peru Basin, leading to different patterns in the solid phase. Even though the same incorporation process into the solid-phase takes place in the Peru Basin and the CCZ - two Pacific nodule areas in the focus of investigating mining-related disturbances - the solid-phase REYSN patterns are different due to the different pore-water REYSN patterns. While the same general pattern (HREY enrichment, negative CeSN anomaly, positive YSN anomaly) is observed in all cores in the Peru Basin, they differ from the REYSN pattern observed in the CCZ (MREY enrichment, no or negative CeSN anomaly). The REY are therefore a suitable parameter for the interregional comparison of sediments."

Referee comment P12 L27-28 Is the ambient pore water REY are equivalent to sea-

water, i.e. the REY enter the sediment through diffusion?

Authors' response: No, the ambient pore water REY are not necessarily the same as the seawater. It is just similar in the Peru Basin. We cannot clearly show which solid phase(s) release they REY to the pore water in the Peru Basin. In other areas, pore water REYSN patterns look very different from the seawater pattern, therefore cycling between solid phase and pore water most likely determines the pore water REYSN pattern.

Referee comment P13 L23 "Both cores, DEA Black Patch and Reference East, are located": : : ! Both DEA Black Patch and Reference East are located: : :" In the preceding paragraph, only Reference East was discussed. The way it is written, it sounds like both cores were just discussed.

Authors' response: Rephrased:

Authors' changes in manuscript: "The Reference East core, as well as the DEA Black Patch core, . . ."

Referee comment P13 L27-29 It looks to me like Reference East is almost certainly anoxic. Nitrate is consumed at a shallow depth and this site has the highest concentrations of dissolved Mn in the deep sediment. It is totally possible for sediments with a lower POC content to be anoxic. Could these trace metal content peaks in Reference East be due to a buried nodule-rich layer that is dissolving, as you have suggested elsewhere?

Authors' response: The Reference East core is not green throughout and nitrate is slightly elevated at depth again. H2S, another indicator for anoxic conditions has never been detected in the Peru Basin (detection limit ca. 0.2 $\mu$mol/L). We would therefore like to stick with our explanation that the sediments are not anoxic here. The trace metals that are released are not typical for Mn nodules, where we would expect the release of Mn, Co (and Fe) and the form of the layers is not comparable to the dissolving

nodules in this and other cores.

Referee comment P13 L35 "They get preserved" revise to "They are preserved"

Authors' response: Changed accordingly.

Referee comment P13 L37 Is there a reference for the claim that turbidites are not common in the area?

Authors' response: We provided two references that similar peaks in the metal concentrations have not been attributed to turbidites in the Peru Basin previously, but rather to the oscillation of the oxic/suboxic boundary.

Authors' changes in manuscript: Added: "In the Peru Basin, solid phase peaks of Cd, Cu, and V have been attributed to the downward progression of the oxic/suboxic boundary during glacial/interglacial cycles which is slowed down by the reactive Fe(II) layer in the clay minerals, and where this oxic front reaches the reactive Fe(II) layer, heavy metals such as V and Cu can be precipitated (authigenic precipitation of U, V, Cu) (König et al., 2001; Koschinsky, 2001)."

Conclusions Referee comment P14 L16 With respect to deep-sea mining, the results show, how variable: : :" → "With respect to deep-sea mining, the results show how variable: : :" Incorrect comma usage.

Authors' response: Comma deleted.

Referee comment P14 L23 Again, what are the halos?

Authors' response: The "halos" are the brown layers surrounding the buried nodules. This was described in section 3.1 core descriptions. Halos form when the nodules oxidize the surrounding suboxic sediment. Written more specifically in section 4.2.

Authors' changes in manuscript: Used to read: "Green sediment gets oxidized 'back' and is tan colored again, as Fe(II) in nontronite is oxidized to Fe(III) (König et al., 1997; Russell et al., 1979)..." Now reads: "Fe(II)-rich sediment gets oxidized 'back' and is

tan colored again (the 'halo'), as Fe(II) in nontronite is oxidized to Fe(III) (Dong et al., 2009; König et al., 1997; Russell et al., 1979)..."

Referee comment P13 (P14?) L27 How exactly can the influence of dissolving nodules be distinguished from hydrothermal input? Maybe with REY or trace metal ratios?

Authors' response: The nodules are another possibility of metal input into the sediments in nodule areas. The REY can be used to confirm or exclude high temperature (ca. >250°C) hydrothermal activity because under high-temperature hydrothermal conditions, a EuSN-anomaly would be visible in the REYSN pattern.

Authors' changes in manuscript: Added in the discussion, section 4.2 : "The lack of high-temperature hydrothermal influence is also shown in the sedimentary REYSN patterns, which lack an EuSN anomaly, a typical sign of high-temperature hydrothermally impacted sediments (Bau, 1991; German et al., 1990; Michard, 1989)."

Referee comment General Conclusions I would prefer a more thorough discussion of the differences or similarities between the DISCOL and undisturbed sites in the Conclusions (if the frame of the paper is changed as I suggested above).

Authors' response: As written throughout the responses, we did not want to focus more on the differences and similarities between the DEA and reference cores, as this is not possible with the GC data and beyond the scope of this paper.

Referee comment General Conclusions The discussion of the effects of the nodules on local trace metal contents should be more fleshed out here, as well. That is a particularly interesting finding of this study, in my opinion, and worth highlighting more specifically here. For example, instead of generally noting "significant small-scale differences in the mineralogical and chemical composition of sediment cores" in the final paragraph, the specific differences (enrichments in solid and pore water trace metals, difference REY signatures, etc) can be re-stated and summarized here.

Authors' response: We thank the reviewer for this comment, but would like to point out

that a lot of information about the specifics of the impact of dissolving nodules was already included in the conclusions, e.g., the higher Fe/Al ratio in the solid phase and the increased Mn and Co concentrations in the pore water. We rephrased the sentence about the small-scale differences to highlight the connection to the specific samples.

Authors' changes to the manuscript: Used to read: "These dissolving nodules can also lead to significant small-scale differences in the mineralogical and chemical composition of sediment cores..." Now reads: "These dissolving nodules can therefore lead to significant small-scale differences in the mineralogical and chemical composition of sediment cores..."

Referee comment General Conclusions Please use consistent markers for each core in all figures. For example, sometimes Reference East is represented by an empty triangle, sometimes by a filled triangle. Also, I recommend using different colors for each site, rather than shades of gray and green.

Authors' response: We thank the reviewer for this comment but would like to point out, that we specifically chose these symbols for consistency: we always use the same symbols for each core, but the filled or partially filled symbol represent solid phase and the open symbols represent pore water. We think this color scheme helps to draw attention to one of the main differentiations between the cores of the paper – cores with green layers (and the associated processes) and cores without green layers. This is also in line with a comment from reviewer #1, who liked the color scheme in Figure 9.